# Splicing QTL mapping in stimulated macrophages associates low-usage splice junctions with immune-mediated disease risk

Omar El Garwany [1], Nikolaos I. Panousis [1], Andrew Knights [1], Natsuhiko Kumasaka [1], Maria Imaz [1], Lorena Boquete Vilarino [1,2], Anthi Tsingene[1], Alex Tokolyi [1], Cristina Cotobal Martin [1], Tobi Alegbe [1], Monika Krzak[1], Tim Raine [3], Alice Barnett [1], Celine Gomez[1], Daniel J. Gaffney [1,4] ✉ & Carl A. Anderson [1,4] ✉

The majority of immune-mediated disease (IMD) risk loci are located in non-coding regions of the genome, making it difficult to decipher their functional effects in relevant physiological contexts. To assess the extent to which alternative splicing contributes to IMD risk, we mapped genetic variants associated with alternative splicing (splicing quantitative trait loci or sQTL) in macrophages exposed to a wide range of environmental stimuli. We found that genes involved in innate immune response pathways undergo extensive differential splicing in response to stimulation and detected significant sQTL effects for over 5734 genes across all stimulation conditions. We colocalised sQTL signals for over 700 genes with IMD-associated risk loci from 22 IMDs with high confidence (PP4 ≥ 0.75). Approximately half of the colocalisations implicate lowly-used splice junctions (mean usage ratio <0.1). Finally, we demonstrate how an inflammatory bowel disease (IBD) risk allele increases the usage of a lowly-used isoform of *PTPN2*, a negative regulator of inflammation. Together, our findings highlight the role alternative splicing plays in IMD risk, and suggest that lowly-used splicing events significantly contribute to complex disease risk.

Genome wide association studies (GWAS) have uncovered thousands of genetic loci associated with susceptibility to immune-mediated diseases (IMD). Over 90% of these loci are located in non-coding regions of the genome[1], making it difficult to gain insights into causal disease biology. These non-coding disease-associated loci are enriched in gene regulatory regions and are therefore thought to modulate gene expression[2].

Expression quantitative trait loci (eQTL) mapping has been widely used to characterise the downstream effects of genetic variants on gene expression[3,4]. Despite the increasing number of available eQTL datasets[5], IMD-associated loci have remained largely unexplained by existing QTL maps. For example, Chun et al. 2017 (ref. 6) found that only 25% of IMD-associated loci colocalised with eQTLs from three immune cell types.

[1]Wellcome Sanger Institute, Wellcome Genome Campus, Hinxton CB10 1RQ, UK. [2]Centro de Fabricación de Terapias Avanzadas, Monte da Condesa s/n, Campus Vida 15782 Santiago de Compostela, A Coruña, Spain. [3]Department of Gastroenterology, Addenbrooke's Hospital, Cambridge University Teaching Hospitals, Cambridge CB2 0QQ, UK. [4]These authors jointly supervised this work: Daniel J Gaffney, Carl A. Anderson. ✉e-mail: daniel.gaffney@gmail.com; ca3@sanger.ac.uk

Multiple explanations have been put forward to justify the incomplete overlap between GWAS loci and existing eQTL maps, including a need for more diverse molecular QTL maps across disease-relevant cell types and environmental conditions. Moreover, it has recently been suggested that common variants driving complex diseases and gene expression are systematically different, and that alternative molecular QTLs (such as those affecting splicing, chromatin accessibility and chromatin interactions) may be more likely to colocalise with disease-associated loci[7]. Unfortunately, most QTL mapping studies have focussed on associating genetic variation with overall levels of gene expression without considering, for example, variation in transcript isoforms. The few studies that have mapped genetic variants associated with alternative splicing (splicing quantitative trait loci or sQTLs) have shown their promise for understanding disease[8,9]. There is thus an urgent need for sQTL maps to be constructed across a broad range of environmental contexts for disease relevant cell types.

Previous work by Nédélec et al. (ref. [10]) has revealed ancestry-specific regulation of gene expression in macrophages in response to Listeria and Salmonella. Moreover, Rotival et al. (ref. [11]) mapped sQTLs in macrophages exposed to four bacterial stimuli and found over 1400 sQTLs. They highlighted an enrichment of disease-associated loci among response sQTLs, but they have not systematically linked inherited changes in alternative splicing to disease-associated loci. Similarly, Rotival et al. have profiled the alternative splicing landscape of monocytes exposed to four stimuli, but have not robustly linked sQTLs to disease-associated loci. Here, we mapped sQTLs in iPSC-derived macrophages in 12 different cellular conditions obtained from 209 individuals at two timepoints after stimulation. We quantify the extent to which alternative splicing responds to macrophage stimulation, and how sQTLs and response sQTLs (re-sQTLs) are shared across environmental contexts. We also explore the contribution of alternative splicing and sQTLs to IMD risk. Finally, we contextualise our findings within an ongoing scientific debate about the functional and evolutionary relevance of low-usage splicing events and discuss the implications of our work on the design of future transcriptomics studies. Compared to previous work, our work expands on the number of environmental conditions and takes on a disease-oriented approach to understand how disease-associated loci impact alternative splicing in macrophages. We have made our data publicly available via an interactive portal (macromapqtl.org.uk).

## Results

### Macrophage stimulation initiates an alternative splicing programme in innate immune response genes

Induced pluripotent stem cell lines (iPSC) from 209 healthy unrelated individuals, generated as part of the HipSci project[12], were differentiated into iPSC-derived macrophages (Fig. 1a; experimental protocol described in Supplementary Note 1). Major and minor spliceosomal genes were not differentially expressed between iPSC-derived and monocyte derived macrophages ($SRSF2$ had the largest fold change log$_2$FC = 0.79; Data from Alasoo et al. 2015; ref. [13]). RNA was harvested from macrophage precursors at day 0 (Prec_D0) and day 2 (Prec_D2). Naïve macrophages were exposed to a panel of 10 stimuli, and RNA was obtained and processed 6 and 24 h after stimulation (in addition to unstimulated controls; Ctrl_6 and Ctrl_24), resulting in a total of 24 different conditions. Our stimulants were selected to cover a wide range of innate immune exposures. This includes pro- and anti-inflammatory cytokines (IFNβ/IFNB, IFNγ/IFNG, interleukin-4/IL4), synthetic viral mimics (Resiquimod/R848, Poly I:C/PIC), bacterial mimics Pam3CSK4/P3C and LPS with and without other inflammatory cytokines (sLPS, CD40 ligand + IFNγ + sLPS/CIL, interleukin-10 + sLPS/LIL10), and myelin basic protein (MBP) to mimic brain-resident macrophage response to stimulation (Supplementary Data 1). In total, this resulted in 4698 RNA-seq libraries across all 24 conditions (Supplementary Data 2).

We quantified alternative splicing from split reads (reads mapping across two splice junctions) using Leafcutter, which quantifies alternative splicing as intron usage ratios[14]. We derived an intron usage ratio matrix for each of the 24 conditions and used this for differential splicing analysis and sQTL mapping (Fig. 1b).

Macrophage precursors (Prec_D0) clustered separately from all other conditions in a UMAP projection of intron usage ratios across conditions (Fig. 2a, b) (Pearson correlation coefficient between Prec_D0/Prec_D2 = 0.74). Precursors at day 2 (Prec_D2) clustered together with the fully differentiated conditions, suggesting that macrophage splicing programmes are activated early in the seven day long differentiation process (Supplementary Note 1) (Pearson correlation coefficient between Prec_D2/Ctrl_6 = 0.88 and Prec_D2/Ctrl_24 = 0.92; Supplementary Fig. 1). We also observed a clear separation between cells harvested after 6 and 24 h, an effect that was not clearly detected for unstimulated macrophages (Ctrl_6 and Ctrl_24; Fig. 2a, b). These findings demonstrate that both differentiation of iPSCs into macrophages and macrophage stimulation initiate profound and temporal alternative splicing changes.

To identify genes that are differentially spliced upon immune stimulation, we undertook differential splicing analysis using Leaf-Cutter (Methods). We identified a total of 3464 genes with altered splicing in stimulated versus unstimulated macrophages after either 6 or 24 h (False discovery rate adjusted Leafcutter $P$-value < 0.05 and absolute log effect size > 0.5). Macrophages stimulated with IL4 had the fewest differentially spliced genes (110 and 94 genes after 6 and 24 h, respectively; Fig. 2c), in line with previous reports that showed stimulating macrophages with IL4 had little impact on splicing[15].

We next undertook pathway enrichment analysis to identify REACTOME pathways enriched with differentially spliced genes following stimulation using Enrichr[16]. At a high-level, enriched pathways re-capitulated known macrophage responses to pathogens, including enrichments of cytokine signalling, membrane trafficking, TCA cycle and metabolism of amino acids, response to stress and apoptosis pathways (Supplementary Figs. 2 and 3 and Supplementary Note 2). Six hours after bacterial stimulation with LPS (sLPS_6), differentially spliced genes were enriched for pathways relevant to macrophage response, including "Cytokine Signalling" (84 genes; Enrichr $P$-value = $9.1 \times 10^{-8}$), "Vesicle-mediated transport" (68 genes; Enrichr $P$-value = $10^{-6}$) and "Class I MHC mediated antigen processing and presentation" (47 genes; Enrichr $P$-value = $7.95 \times 10^{-6}$) (Supplementary Data 3). Several genes coding for members of the RAB family of proteins were enriched within the vesicle-mediated transport pathway, which is also known to be regulated via alternative splicing[17] (including $RAB1B$, $RAB4A$, $RAB9A$, $RAB11A$, $RAB13$, $RAB14$ and $RABEPK$; Fig. 2d, Supplementary Data 3). RABs are GTPases that regulate membrane trafficking by controlling the formation, movement and recycling of vesicles[18]. For example, we observed greater usage of the first exon of a non-canonical transcript of $RAB13$ ($RAB13$-205), coupled with lower usage of the canonical first exon ($RAB13$-201) (Difference in Percentage Spliced In ($\Delta$PSI) = 0.046 and −0.05, respectively). The two annotated transcripts $RAB13$-201 and $RAB13$-205 result in protein products of 203 and 122 amino acids respectively, which indicates that a different $RAB13$ protein isoform is produced following stimulation with LPS.

Our stimulation panel also included viral stimuli, which enabled us to explore if viral response genes were differentially spliced upon stimulation. We observed that six hours after stimulation with dsRNA viral mimic PolyI:C (PIC_6), genes involved in viral sensing and RIG-I/MDA5-mediated activation of the antiviral cytokine Interferon β (IFNβ) are differentially spliced (Supplementary Data 3). RIG-I/MDA5 belong to the RIG-I-like family of receptors which sense dsRNA and activate type I interferons in response (e.g. IFNβ)[19,20]. Differentially spliced genes include genes coding for the RIG-I and MDA5 receptors themselves ($DDX58$ and $IFIH1$ respectively), but also genes involved in IFNβ regulation, such as $TRIM25$[21], $TANK$[22], and $RIPK1$[23]. These findings are

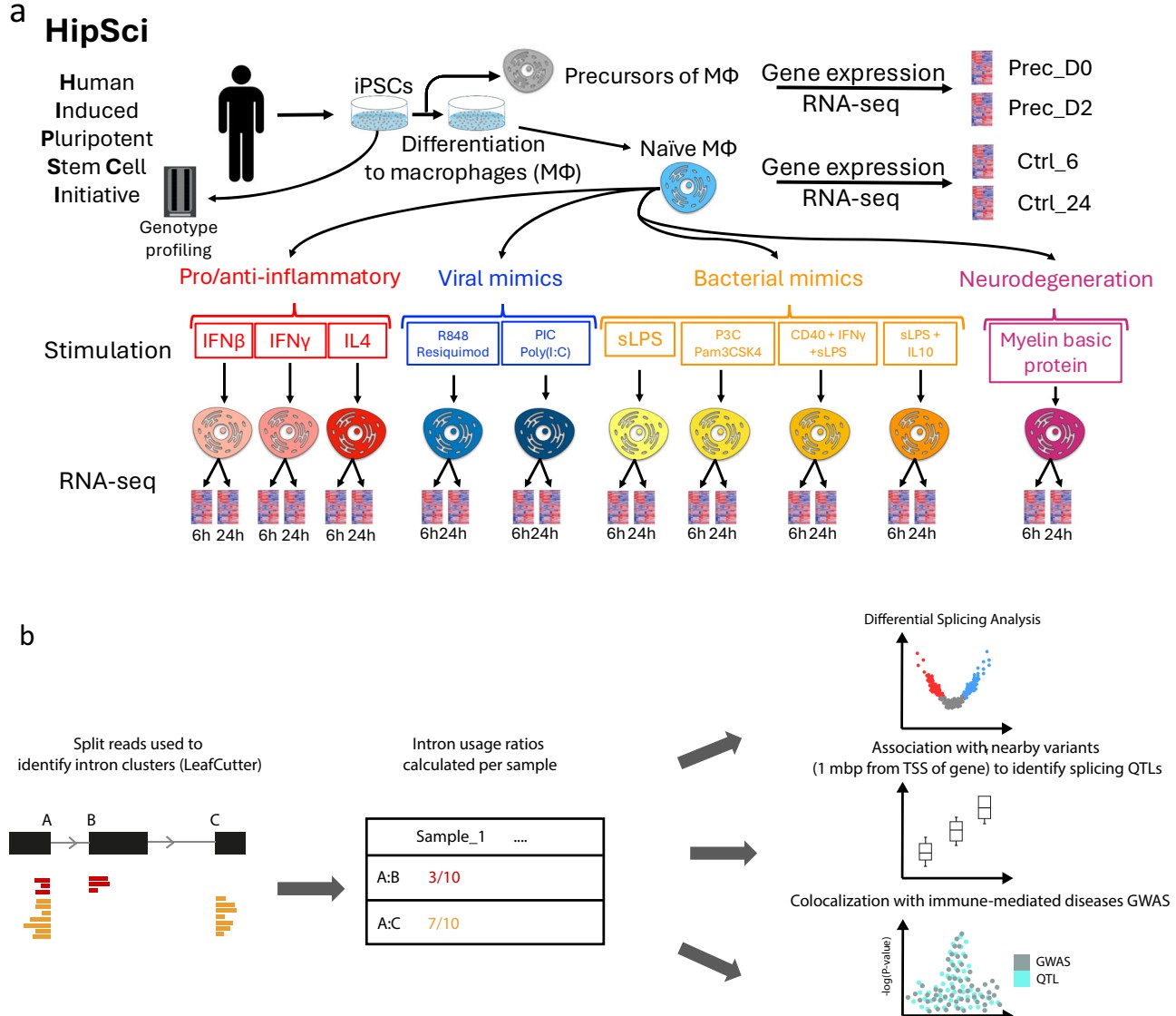

**Fig. 1 | Overview of MacroMap.** Overview of study: **a** Genotyped iPSC cell lines were differentiated into macrophages, and RNA was harvested before differentiation (Prec_D0) and 2 days after starting differentiation (Prec_D2). RNA was also harvested from differentiated macrophages at 6 and 24 h (Ctrl_6 and Ctrl_24). Naïve macrophages were then exposed to a panel of 10 stimuli and RNA was harvested at 6 and 24 h after stimulation. **b** Split reads were used to quantify intron usage ratios on an individual level using LeafCutter. Split reads were then used for differential splicing analysis between naive and stimulated conditions, and as a quantitative trait to map splicing quantitative trait loci (sQTLs). sQTLs were then colocalised with 22 immune-mediated disease GWAS summary statistics.

supported by previous work showing that regulatory components of the RIG-I/MDA5-mediated IFNβ activation pathway have different isoforms that modify antiviral response[24–26].

These findings demonstrate that iPSC derived macrophages can faithfully recapitulate known biological pathways activated by different stimuli. This motivated us to investigate how genetic variation affects alternative splicing in macrophages and how such genetic variation may, in turn, predispose to IMDs.

**Macrophage stimulation increases the number of genes with significant sQTL effects**

To understand which alternative splicing events were affected by genetic variation, we mapped splicing QTLs using intron usage ratios as a quantitative trait. Intron usage ratios were normalised (quantile normalised and rank-based inverse normal transformed) and introns with low variance (standard deviation <0.005), or intron clusters without split reads in more than 40% of samples, were excluded (Methods). We identified a median of 82,058 introns (75,987–105,841

introns across conditions with the greatest number of introns seen in Prec_D0; Supplementary Fig. 4). Each gene had a median of 7 introns and up to 10,851 genes were quantified per condition.

We mapped sQTLs within a ± 1 Mbp window centred around the transcription start site (TSS) of each gene. We noted both high replication rates and high correlation between effect size estimates for our sQTLs and those from two macrophage datasets from the eQTL Catalogue ($\pi_1 > 0.7$; pearson correlation coefficient of effect sizes $\rho > 0.9$; Supplementary Fig. 5). We also used multivariate adaptive shrinkage (mash[27]) to compare sQTL effect sizes between naive and stimulated conditions (we used Ctrl_24 and Ctrl_6 as our baseline conditions). Mash reports a significance measure known as local false sign rate (LFSR; Methods), which we used to identify sQTLs whose effect sizes change significantly upon stimulation (response sQTLs).

We called significant sQTLs at a false discovery rate (FDR) ≤ 0.05. Across all conditions we detected a total of 5734 sGenes (median number of sGenes per condition = 1580 and Prec_D2 had the most sGenes = 1881) (Fig. 3a, b). Of these, 875 sGenes (15.2%) had at least one

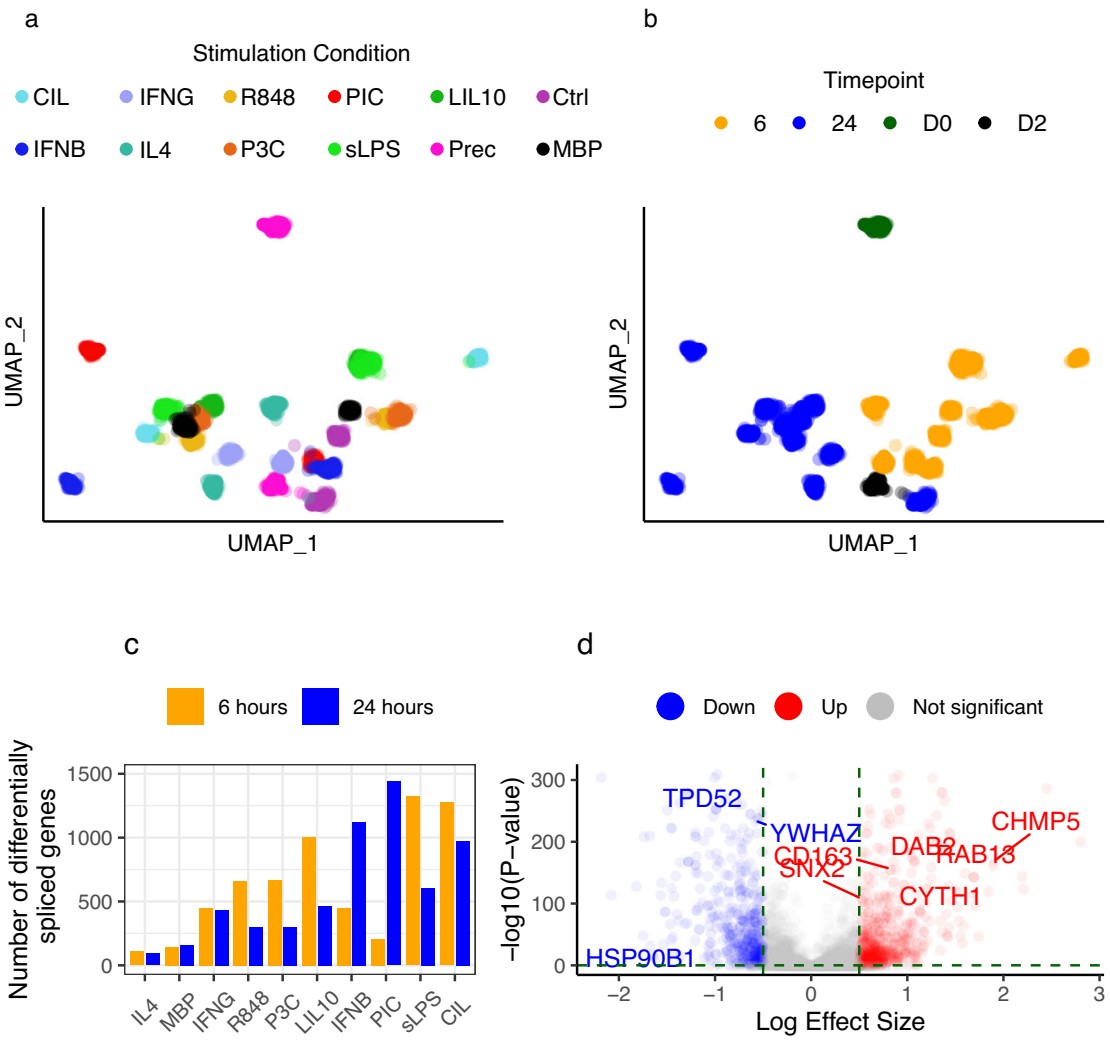

**Fig. 2 | Uniform Manifold Approximation and Projection of all MacroMap conditions and differential splicing analysis results.** UMAP of intron usage ratios in different stimulation conditions, coloured both by (**a**) different stimulation conditions and (**b**) by time point. **c** Number of differentially spliced genes between naive and stimulated macrophages after 6 h (yellow) and 24 h (blue). **d** Volcano plot showing differentially spliced genes 6 h after sLPS stimulation, with log effect size on the x-axis (for each gene the intron with the largest absolute effect size is shown) and -log$_{10}$ of adjusted *P*-value on the y-axis (Leafcutter *P*-values were FDR-adjusted). Colours indicate the direction of intron usage change (blue indicating reduced usage and red indicating greater usage in stimulated cells versus naive cells). Genes that belong to the "vesicle-mediated transport" REACTOME pathway are indicated. Prec: iPSC precursor, Ctrl: Control, sLPS: Lippolysaccharide, IL4: Interleukin 4, IFNG: Interferon γ, IFNB: Interferon β, CIL: CD40 + IFNG + sLPS, R848: Resiquimod, PIC: Polyl:C, LIL10: sLPS + Interleukin 10, P3C: Pam3CSK4, MBP: Myelin Basic Protein. D0: Day 0 of iPSC differentiation, D2: Day 2 of iPSC differentiation, 6: 6 h following stimulation, 24: 24 h following stimulation.

response sQTL (LFSR < 0.05). We then asked which of the two stimulation timepoints (6 and 24 h) was more likely to have response sQTL effects across stimulation conditions. For viral mimic PIC, and pro-inflammatory stimuli IFNG and IFNB, we found a larger proportion of sGenes with a response sQTL at 24 h only than 6 h only (up to 55.4% of all response sQTLs versus up to 11.8%, respectively; Fig. 3c). For all other conditions, we found a larger proportion sGenes with a response sQTL at 6 h only than 24 h only (up to 50.2% of all response sQTLs versus up to 34.2%, respectively; Fig. 3c). This is in line with previous reports that found a rapid short-lived macrophage response to LPS stimulation and a slow long-lasting macrophage response to PIC stimulation[28].

Similar to previous reports[29], we found that lead sQTL SNPs are located closer to intron boundaries than to the TSS of their genes. On average per condition, 24.5% of sGenes had a lead SNP within 10 kbp of their TSS, compared to 46.8% within 10 kbp of either the 5' or 3' intron boundaries (Supplementary Fig. 6). Previous sQTL mapping efforts have also shown that sQTL signals are largely independent from eQTL

signals, and therefore overall levels of gene expression and alternative splicing are genetically regulated via distinct mechanisms[9,30,31]. To verify this, we performed statistical colocalisation[32] (Methods) between sQTLs and eQTLs derived from the same data[33]. We found that, on average across conditions, only 25% of sGenes likely share a single causal variant with an eGene in the same condition, even when applying a permissive colocalisation cutoff (PP4 ≥ 0.1; Supplementary Fig. 7; Methods), indicating that the majority of sQTL signals are largely independent from eQTL signals. We therefore hypothesised that colocalisation of sQTLs with disease-associated loci may identify additional disease effector genes distinct from those already implicated via eQTLs.

**Splicing QTLs identify additional GWAS loci effector genes not detected by expression QTLs**

Macrophage sQTL maps can also be used to build hypotheses about how IMD risk loci confer risk by dysregulating alternative splicing. To

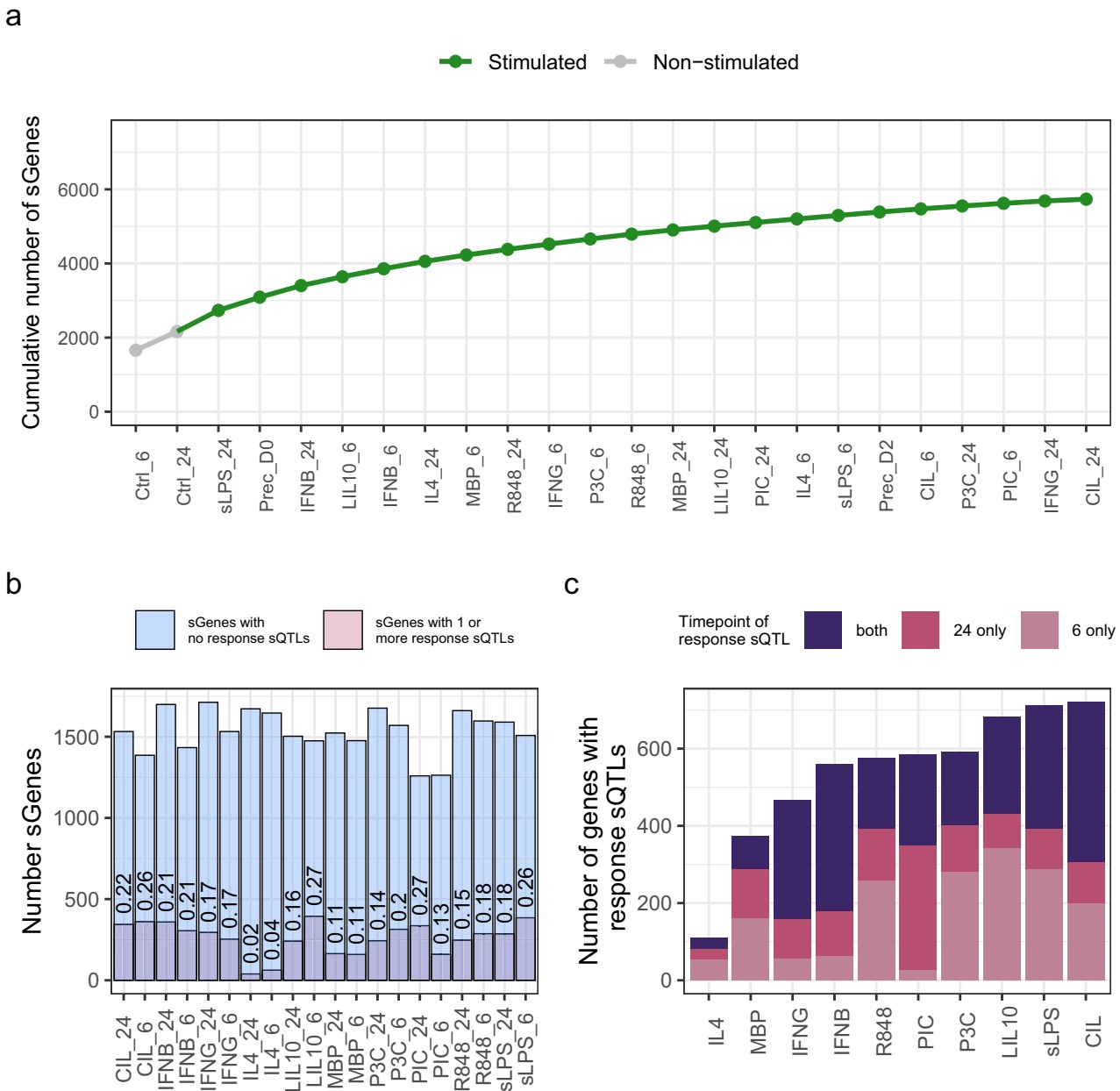

**Fig. 3 | Significant sQTLs show stimulation-specificity. a** Cumulative number of genes with significant splicing QTL effects, with unstimulated conditions indicated in grey. **b** Total number of significant sQTLs per condition and proportion of response sQTLs within each condition (sQTLs with LFSR < 0.05; Methods). **c** Number of genes, per condition, with at least one response sQTL at 6 h, 24 h or both.

this end, we used statistical colocalisation (using coloc; Methods) between sQTL and GWAS association signals to quantify the probability of a sQTL sharing a causal variant with genetic association signals from 22 IMD GWAS summary statistics (downloaded from GWAS catalogue[34]; Methods and Supplementary Data 4).

Across all 22 IMDs, we identified 715 unique genes (1344 introns) with an sQTL signal in at least one condition that likely shares a causal variant with an IMD risk locus (PP4 ≥ 0.75; Fig. 4a). On average, 68% of sQTL-disease colocalisation events could only be detected in a single condition (Supplementary Fig. 8). However, a smaller percentage of colocalised sQTLs (25%) showed evidence of being response sQTLs (LFSR < 0.05; Supplementary Fig. 9). This suggests that profiling diverse stimulation conditions may be necessary to confidently colocalise disease-associated loci with sQTLs, but a smaller subset of colocalised sQTLs are likely to be true response sQTLs. Additionally, based on hierarchical clustering of

LFSR values, IL4_6 and IL4_24 had the fewest response sQTLs that colocalised with GWAS signals, while sLPS_6, CIL_6 and LIL10_6 (all stimulated with LPS) yielded the most, recapitulating results from the differential splicing analysis (Fig. 4b).

We then compared how many tested loci colocalised with each type of molecular QTL and found that 52.6% (820/1558) of tested loci were likely to share a single causal variant with either an eQTL, sQTL or both. This colocalisation yield was higher in IMDs than in a set of 11 psychiatric and cognitive traits (34.7%; Supplementary Fig. 10). Approximately half of the colocalised loci (312 loci or 20.1% of tested loci) colocalised solely with an sQTL (sQTL PP4 ≥ 0.75 > eQTL PP4). Conversely, 9% of tested loci colocalised solely with an eQTL (eQTL PP4 ≥ 0.75 > sQTL PP4; Fig. 4c). The contribution of loci that colocalised solely with an sQTL remained larger than the contribution of loci that colocalised solely with an eQTL across a range of exclusion PP4 cutoffs (Supplementary Fig. 11). This clearly demonstrates both the

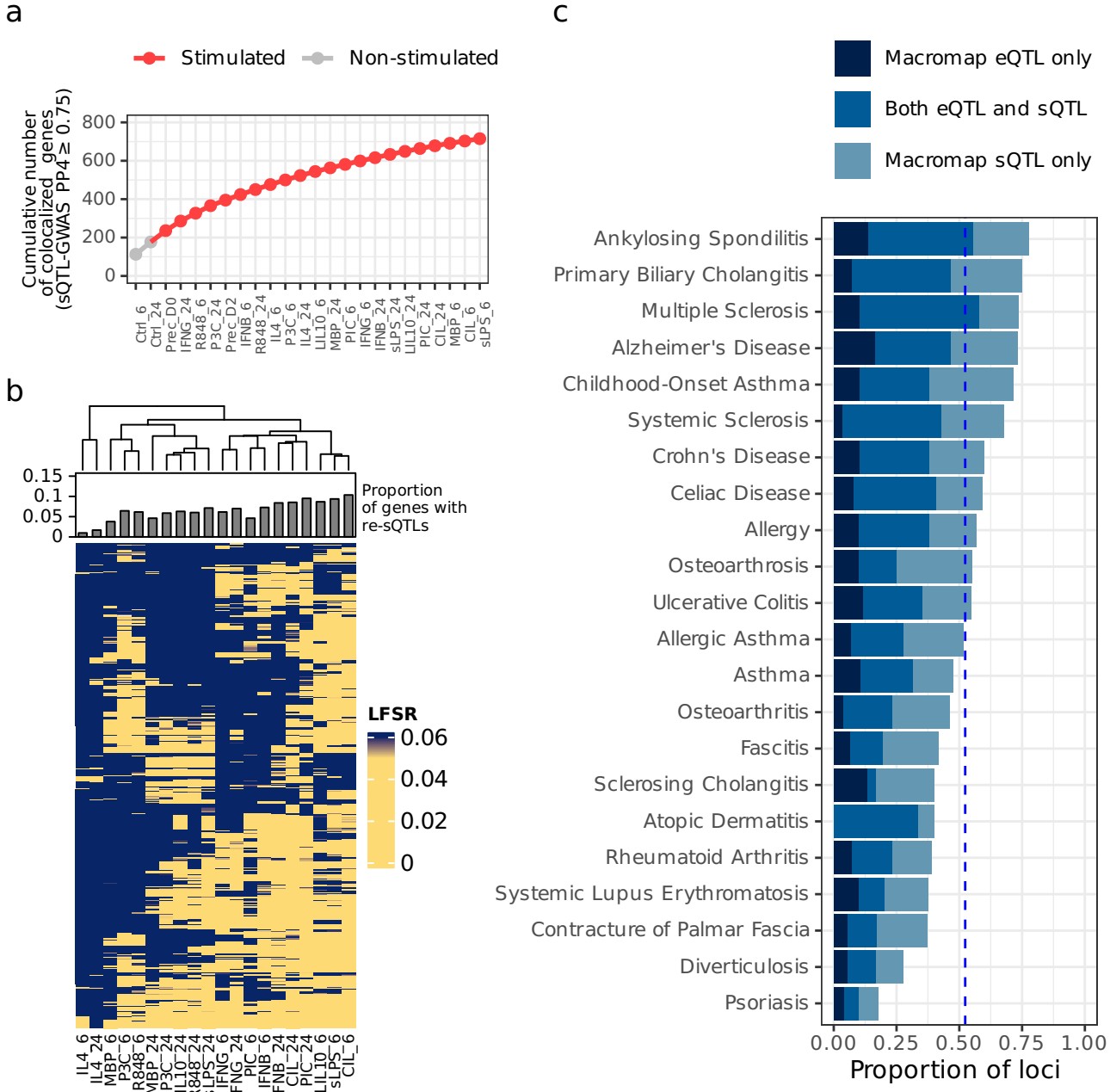

**Fig. 4 | Colocalisation of sQTLs with immune-mediated disease risk loci.**
**a** Cumulative number of genes with GWAS-sQTL colocalisations (PP4 ≥ 0.75) across different conditions, with unstimulated conditions shown in grey on the left.
**b** Heatmap and hierarchical clustering of LFSR values for all the colocalised sQTL effects (PP4 ≥ 0.75) across all 22 IMDs. On top of the heatmap is a barplot showing the proportion of colocalised sQTL effects that are response sQTLs (LFSR < 0.05).
**c** Proportion of genome-wide significant loci that share a single causal variant (PP4 ≥ 0.75) with an eQTL only, an sQTL only or both.

value of sQTLs for identifying GWAS effector genes and the important role that alternative splicing plays in complex disease risk (Fig. 4c).

**Lowly-used alternative splicing events are associated with complex disease risk**
We next sought to characterise the colocalised sQTL introns, by asking how often colocalised sQTL splicing events are used in observed transcripts. There is ample evidence that lowly-observed aberrant splicing underpins several inherited diseases such as Spinal Muscular Atrophy and Duchenne Muscular Dystrophy[35–37], but it is unclear to what extent lowly-observed splicing events contribute to complex diseases.

We define low-usage introns based on the mean IUR in the genotype group with two copies of the usage-increasing (hereafter referred to as high-genotype IUR). We observed that 43% of colocalised sQTL introns have a high-genotype IUR < 0.1 across samples (Fig. 5a; 43% in non-stimulated conditions and 42.9% in stimulated conditions; Supplementary Fig. 12). Over 96.3% of these introns had non-zero usage in at least 30 RNA-seq samples (Fig. 5b), indicating that these splicing events can be reliably observed in multiple RNA-seq samples and individuals, but with relatively low IUR. Generally, we noted that there was a small but significant difference in absolute effect size between common- and low-usage splice junctions that colocalised with disease associated loci (mean effect size = 0.75 and 0.7 and

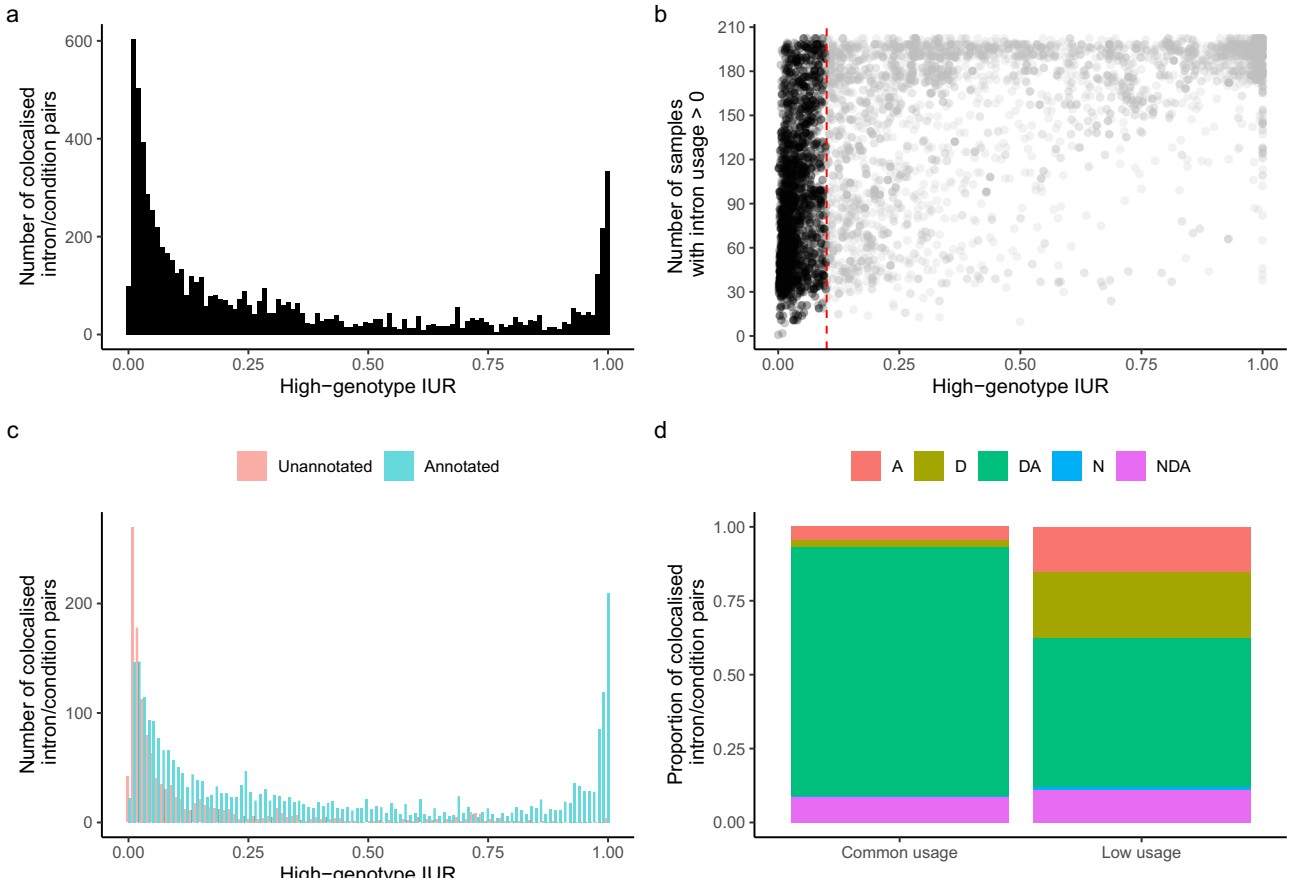

**Fig. 5 | Low-usage introns underping immune-mediated disease risk loci.**
**a** Distribution of high-genotype intron usage ratio (IUR) for colocalised introns, showing a peak close to 0 (**b**) number of samples where each intron is supported by at least one split read (y-axis) shown against the high-genotype IUR of each sample (x-axis). Red vertical line at high-genope IUR = 0.1. **c** Distribution of high-genotype IUR for colocalised introns coloured by annotation in GENCODE v45, showing an enrichment of unannotated introns among introns with high-genotype IUR < 0.1 (**d**) proportion of colocalised splice junctions with either a known acceptor/donor combination (DA), a novel donor (A), a novel acceptor (D) a novel combination of known donor/acceptor splice sites (NDA), or a novel acceptor and donor (N) for colocalised common-usage and colocalised low-usage splice junctions.

standard deviation = 0.27 and 0.27, respectively; Supplementary Fig. 13). However, we did not observe differences in alignment score or multi-mapping rates of split reads between common-usage and low-usage introns (Supplementary Fig. 14, 15). Additionally, 99.3% of low-usage introns featured the canonical GT/AG dinucleotides at their donor and acceptor splice sites (Supplementary Fig. 16).

Interestingly, 49.7% of low-usage introns are not found in any annotated transcripts in GENCODE v45, whereas only 15.9% of introns with high-genotype IUR ≥ 0.1 are absent from GENCODE v45 (Fig. 5c), in line with previous reports showing that low-usage splicing events tend to be unannotated in transcript databases[38,39]. Despite the strong enrichment of unannotated introns in low-usage introns, only 0.8% of low-usage introns had both a novel acceptor and a novel donor donor splice site. 49% of low-usage introns feature either a known acceptor site only, a known donor site only or an unannotated combination of known acceptor and donor sites (Fig. 5d).

We then asked if the set of colocalised introns was enriched or depleted with low-usage introns compared to all tested introns. We found that colocalised introns were generally depleted for low-usage introns, but depletion becomes weaker at the higher end of IUR cutoffs (Fisher's exact test odds ratio = 0.6–0.92 across high-genotype IUR cutoffs = 0.01–0.1; Supplementary Figs. 17 and 18). This suggests that the large proportion of colocalised introns with high-genotype IUR < 0.1 is a result of testing a set of introns with a similarly large proportion of low-usage introns.

In order to validate the replicability of low-usage colocalised introns in large-scale RNA-seq databases, we leveraged Intropolis[38], a database of introns identified using over 21,000 RNA-seq samples from the Sequence Read Archive. We found that only 1.4% of low-usage colocalised introns (N = 10) were not present in Intropolis. Additionally, 92.3% of the low-usage colocalised introns were identified in at least 100 RNA-seq samples (Supplementary Fig. 19).

To further assess the replication of a subset of our low-usage splice junctions, we generated single-cell long-read RNA-seq data. We focussed our validation on a set of 99 low-usage splice junctions that we colocalised with Crohn's disease-associated signals (high-genotype IUR < 0.1 and PP4 ≥ 0.75). The long-read dataset consists of 15 terminal ileum biopsies from eight healthy and seven non-inflamed CD individuals. We previously identified major epithelial and immune cell types from the same samples via short-read single cell RNA-seq[39]. We therefore matched the cell barcodes from the same samples to their cell-type annotation obtained from short-read RNA-seq to identify myeloid cells (Supplementary Fig. 20).

Full-length cDNA was sequenced using Pacbio's concatenation method MAS-seq[40]. We generated a median of 96,347,033 segmented long-reads per sample, with a median read length of 738 basepairs. After quality control (Methods), we obtained a median of 45,451,801 deduplicated long read UMIs per sample and a median of 2627 and 4324 cells per sample for healthy and CD samples respectively (Supplementary Data 5). As the biopsies we used came largely from

individuals with no inflammation, we were able to identify only 1203 myeloid cells across all 15 samples (Supplementary Fig. 20). Despite this small number, we validated the existence of 50.5% of CD-implicated low-usage splice junctions in myeloid cells only (50 splice junctions). Replication rate increased to 76.8% (76 splice junctions) when we performed the replication across all immune cell types (B cells, T cells, Plasma B cells and Myeloid cells; 22,579 cells; see Data Availability for cell-level counts of validated CD splice junctions and their isoforms). 77.6% of replicated splice junctions are detected in at least two samples ($N = 59$) and 82.9% of replicated splice junctions are detected in at least two cell barcodes ($N = 63$; Supplementary Fig. 21). The replicated splice junctions included annotated splice junctions ($N = 38$), splice junctions with an annotated donor only ($N = 14$) or acceptor only ($N = 10$), or a novel combination of a known acceptor and a known donor ($N = 14$), suggesting that the replication reflects both annotated and unannotated splice junctions.

### A rare alternative splicing event likely underpins inflammatory bowel disease risk at the *PTPN2* locus

To demonstrate how low-usage sQTLs can dysregulate alternative splicing and predispose to IMDs, we further investigated a *PTPN2* sQTL that implicates a lowly-used intron that colocalised with an inflammatory bowel disease (IBD) associated risk locus at 18p11.21 (Fig. 6a). Multiple lines of evidence, including coding variants associated with monogenic IBD[41,42] and mouse knock-out models[43,44], have suggested *PTPN2* is the effector gene at 18p11.21, though this remains to be established. It is not yet known if and how common IBD-associated SNPs affect the expression of *PTPN2*. We observed that the lead IBD SNP at 18p11.21 (rs80262450; 18-12818923-G-A) is associated with higher risk of IBD and with increased usage of splice junction chr18:12,817,365-12,818,944 (Fig. 6b, d and Supplementary Fig. 22). rs80262450 is located 21 base pairs downstream of the donor splice site, and is the lead SNP for both the sQTL and IBD association signals, strongly suggesting its involvement in the aberrant splicing event at this locus. The *PTPN2* sQTL signal colocalised with the IBD signal in 13 conditions (with a strict PP4 cutoff $\geq 0.9$; Fig. 6c), but did not colocalise with any eQTLs mapped from the same data[32]. Despite the relative rarity of this splice junction, we detected this colocalisation using GTEx sQTL summary statistics[9], where this sQTL signal was also colocalised with the IBD locus (PP4 $\geq 0.9$) in 14 tissues, including whole blood (Supplementary Data 6 and Supplementary Fig. 23), indicating that this rare splicing event can be reliably detected in a large number of tissues from an independent dataset.

The directions of effects of rs80262450 on intron usage and IBD risk (effect size in sLPS_6 = 1.22 and odds ratio = 1.17, respectively) suggest that an increase in the relative abundance of the splice junction is associated with increased risk of IBD (Fig. 6d). Given that mouse knock-out studies of *PTPN2* suggest the gene plays an anti-inflammatory role in macrophages, we hypothesise that increased usage of the implicated splice junction attenuates the role of *PTPN2* as a negative regulator of inflammation, which in turn increases the risk of IBD. Although the implicated splice junction exists in only one annotated transcript (*PTPN2-205*), we found that it is used in at least four transcripts in a blood sample that was sequenced using Pacbio's long-read sequencing (Supplementary Fig. 24). Therefore, the isoform that underpins the colocalised macrophage sQTL for *PTPN2* remains inconclusive.

### sQTL colocalisations converge on dysregulated pathways in IMDs

IMD-sQTL colocalisations in disease-relevant cell types can reveal how genetic variation dysregulates biological pathways that enable the cell to perform its normal functions.

For example, the IBD-associated locus at 1q31.3 and the ulcerative colitis (UC) associated locus 22q12.3 colocalised with *DENND1B* and *TOM1* sQTLs in 18 and 10 conditions, respectively (with a strict PP4 cutoff

$\geq 0.9$; splice junctions chr1:197,715,074-197,772,868 and chr22:35,333,497-35,334,328, respectively; Supplementary Data 7 and 8; Fig. 7). Both of these genes contribute to the regulation of vesicle trafficking. *DENND1B* is a guanine exchange factor (GEF) that activates several RAB GTPases[45]. RAB GTPases are a diverse set of molecules that regulate different aspects of vesicle-mediated transport, with over 70 RAB GTPases discovered so far[46]. RAB GTPases are activated by GEFs, and subsequently recruit effector proteins, including proteins required for vesicle uncoating, movement, tethering and fusion, which enables cargo trafficking across cell compartments and membranes[15]. DENND1B is known to interact with Rab35. *TOM1* encodes a protein in the VHS domain family and plays a role in endosomal trafficking. Several interactions between TOM1 and other endosomal trafficking effector proteins have been outlined (reviewed in ref. 47). For example, Seet et al.[48] showed that TOM1 is recruited by another protein called endofin to the early endosome, and that TOM1 interacts with the endosomal clathrin coat. However, the function of this interaction remains incompletely understood.

The *DENND1B* sQTLs that colocalise with the IBD-associated signal 1q31.3 suggests that an alternative exon inclusion event may underpin the IBD risk. The risk allele of the lead IBD variant (rs2224873) is associated with increased inclusion of an exon that is skipped in the canonical transcript *DENND1B-211* (Supplementary Fig. 25). The *TOM1* sQTLs that colocalise with the UC-associated signal 22q12.3 suggests that a cryptic splicing event may be associated with protection from UC. The protective allele of the lead UC variant (rs138788) is associated with increased inclusion of a cryptic exon that resides in intron 10-11 of the canonical transcript *TOM1-209*. This is coupled with a decreased frequency of the splice junction between intron 10-11 (Supplementary Fig. 26). Moreover, the alternative allele of rs138788 may lead to the disruption of the acceptor splice site of the cryptic exon as it changes the dinucleotide sequence from AG to AA (Supplementary Fig. 26). Additionally, the two cryptic splicing events that lead to the inclusion of the cryptic exon were validated in the Intropolis database, where both were observed in over 500 RNA-seq samples.

## Discussion

In this work, we mapped splicing QTLs in iPSC-derived macrophages to understand how splicing is genetically controlled in macrophages under 12 different environmental contexts. Our differential splicing analysis reinforces the role of alternative splicing in regulating important innate immunity processes (e.g. vesicle-mediated transport and viral sensing and response), but also highlights the specificity of splice junction usage in response to specific stimuli. Although the role of alternative splicing in response to stimuli has been previously reported[10,11,24,49], our comprehensive and well-powered screen captures pathway-level changes in splice junction usage in response to a broad range of stimuli. Additionally, we found thousands of genes with significant sQTL effects across our array of conditions, and that a considerable proportion of these genes have different effect sizes upon stimulation and were thus defined as response sQTLs.

The primary motivation behind several QTL mapping efforts is to understand the transcriptomic consequences of disease-associated genetic variation. Recently, Mostafavi et al. (ref. 7) suggested that eQTL and GWAS studies are powered to discover systematically different types of variants, and that this may partially explain the limited colocalisation between eQTLs and GWAS risk loci. More than half of the IMD-associated risk loci that we tested likely share a single causal variant with either an eQTL or sQTL, with sQTLs solely contributing 20% of these loci, which clearly demonstrates the added value of sQTLs. This echoes previous work that showed the promise of sQTLs in closing the colocalisation gap[9,15,49]. Mostafavi et al. proposed that systematic differences between GWAS and eQTL association signals are behind this colocalisation gap. Along the same lines, we attribute the large number of sQTL colocalisations to three important features of our discovered sQTLs that make them likely to colocalise with GWAS

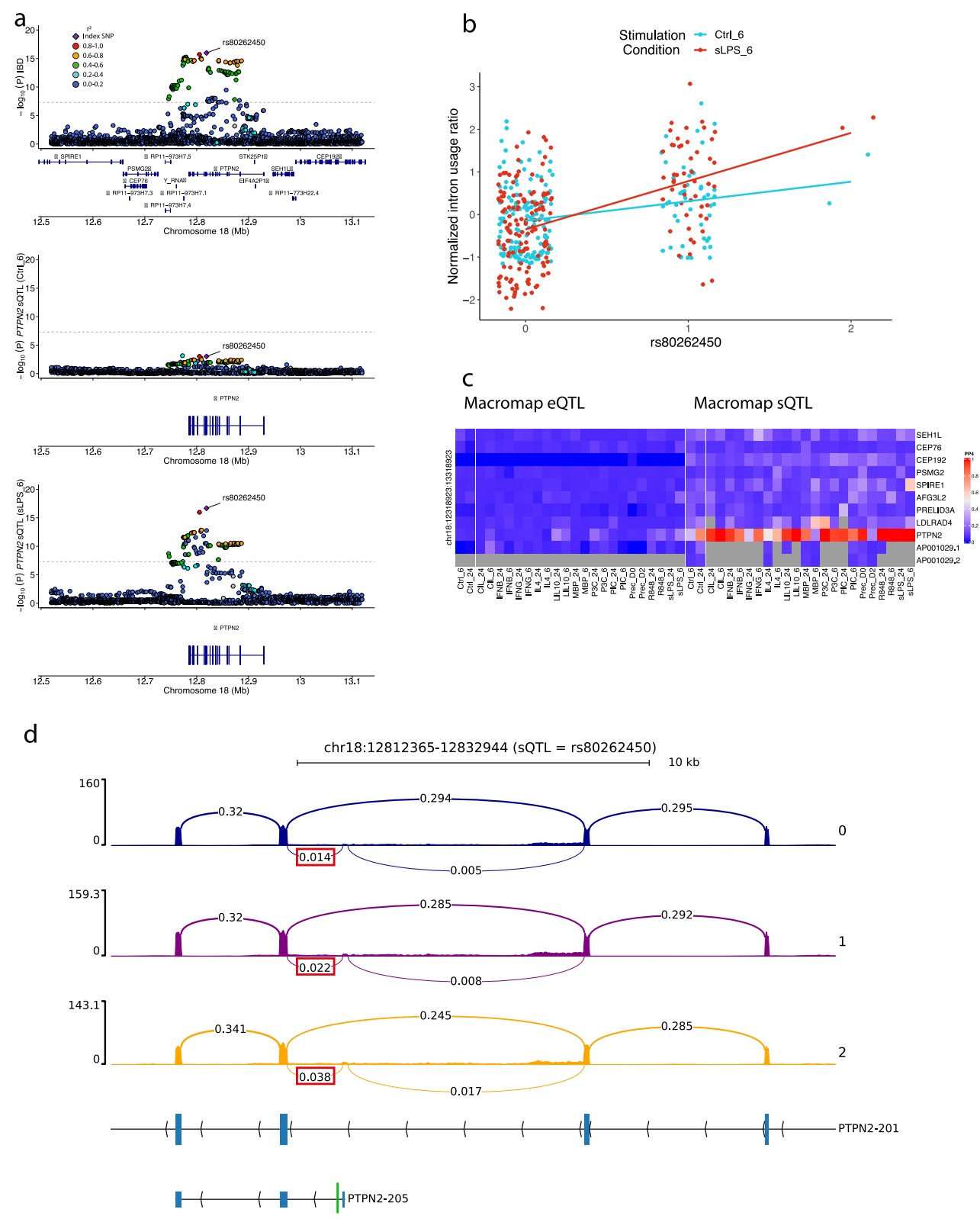

association signals. First, unlike eQTL variants, sQTL variants tend to be located around intron splice sites, rather than the canonical TSS of genes (Supplementary Fig. 6), closer to intronic GWAS association signals that do not colocalise with eQTL signals. Indeed, 63.4% of the lead SNPs in the GWAS loci that we tested were intronic variants. Second, sQTL signals largely do not colocalise with eQTL signals, suggesting that they are driven by distinct genetic associations

(Supplementary Fig. 7). Third, we show that a considerable number of colocalisation events implicate unannotated introns that are lowly used across transcripts. These subtle changes in intron usage are unlikely to be reflected in overall levels of gene expression levels and may therefore remain undetected in eQTL studies.

Currently, it is unclear whether lowly-used splicing events are simply splicing errors[50,51] or if they have functional consequences[11,52].

**Fig. 6 | Example of colocalisation between an IBD risk locus at 18p11.21 and an sQTL for *PTPN2*. a** GWAS regional association plots of the IBD association signal (top), and sQTL association signal for the splice junction chr18:12,817,365-chr18:12,818,944 in unstimulated macrophages (Ctrl_6; middle) and macrophages stimulated with sLPS after 6 h (sLPS_6; bottom). Lead IBD SNP rs80262450 is indicated with an arrow. Colours represent linkage disequillibrium with the index SNP. **b** Normalised intron usage ratios of different genotypes of the lead IBD SNP rs80262450 in Ctrl_6 and sLPS_6, (**c**) heatmap showing evidence of colocalisation (PP4) between the IBD association signal at 18p11.21 and all macrophage eQTLs/sQTLs in the locus (in all conditions), (**d**) RNA-seq coverage of the intron cluster

where the *PTPN2* sQTL effect is detected in sLPS_6 stratified by the number of copies of the alternative allele of rs80262450. Bars represent the number of reads and arcs represent the usage of different introns (only five splice junction ratios are shown for clarity and the colocalised sQTL splice junction is indicated in a red box). Canonical transcript *PTPN2-201* and non-canonical transcript *PTPN2-205* (the only annotated transcript with the implicated splice junction) are shown underneath, with blue boxes representing exons and the position of rs80262450 on *PTPN2-205* is shown by the green line. A similar RNA-seq coverage plot in unstimulated macrophages (Ctrl_6) is shown in Supplementary Fig. 21.

For example, Pickrell et al. 2010 (ref. 50) interpreted the lack of evolutionary conservation around lowly-used splice sites as evidence that they are functionally-irrelevant. This interpretation has been debated over the past decade[53–56]. Here, we show that many of these lowly-used splice junctions may underpin disease-associated genetic effects on alternative splicing, and should not be discarded as noisy and functionally irrelevant without further investigation. The *PTPN2* example is particularly intriguing as the implicated splice junction falls within an Alu element, a class of Short Interspersed Nuclear Elements (SINE; Dfam *E*-value = $4.7 \times 10^{-98}$) that constitute 11% of the human genome[57]. RNA binding proteins normally repress the expression of newly-incorporated Alu elements, but decreased repression over long evolutionary periods provides a substrate for new functions[58,59]. The risk-increasing effect of the lowly-used *PTPN2* splice junction could therefore represent a harmful evolutionary byproduct that attenuates the anti-inflammatory effect of *PTPN2*. This remains to be validated by functional studies that profile the functional consequences of *PTPN2* isoforms that contain this splice junction. With the recent success of RNA therapeutics such as splice-switching antisense oligonucleotides (ASO), it may be possible to "contain" these evolutionary side effects via therapeutic interventions that decrease the proportion of these lowly-used splice junctions[60,61]. This should provide incentive for the complex disease and transcriptomic community to understand and interrogate the functional consequences of the splicing events that underpin complex disease risk.

Although our dataset represents the largest resource for studying how alternative splicing is genetically controlled in iPSC-derived macrophages, we acknowledge three main limitations. First, although macrophages differentiated using our experimental protocol have been shown to be transcriptionally similar to monocyte-derived macrophages[13], they still do not capture the local environment of tissue-resident macrophages. This may limit our ability to understand the effect of local micro-environments on the transcriptome of macrophage[62]. Second, intron usage ratios do not provide information about the relative abundance of full transcripts. For example, we demonstrated the potential effects of splicing events on the canonical transcripts of *PTPN2*, but it is still unclear which transcripts these splicing events may have originated from. Fortunately, long-read sequencing and its algorithms for isoform quantification are becoming increasingly mature[39,63–65]. We therefore expect that a more direct quantification of transcript usage will be attainable, and that it will make alternative splicing a more routine part of transcriptomic profiling studies. Second, similar to other QTL studies, long-range linkage disequillibrium in disease-associated loci makes the identification of effector genes challenging. This is exemplified by our *PTPN2* sQTL example, where an *RP11-973H7.1* eQTL in relevant tissues also colocalises with the IBD-associated signal. This makes it difficult to statistically distinguish which QTL underpins the IBD risk without functional validation (Supplementary Fig. 27). Third, our observation that low-usage splice junctions contribute to IMD risk needs further investigation. It is unclear if these splice junctions themselves are truly functional, or if the IMD risk is a result of reduced usage of the canonical isoforms of genes (i.e. inefficient splicing) which may be reflected in increased usage of low-usage splice junctions. Studies that examine

the fate and role of low-usage splice junction in the context of complex disease are set to improve our understanding of the function consequences of low-usage splice junctions[66].

In summary, our findings highlight an important role for alternative splicing in macrophage response to environmental cues, and that its dysregulation explains a large proportion of IMD-associated susceptibility loci. We anticipate that improved long-read sequencing technologies will facilitate whole isoform quantification in different cellular contexts, which will open the door for a better understanding of the role of different isoforms in innate immune response. Finally, we recommend that alternative splicing quantification should be an integral part of future QTL studies (including single cell studies), and that it will be increasingly relevant to understand the transcriptomic effects of disease-associated risk loci.

## Methods
### Sample collection and ethical approval of Induced Pluripotent Stem Cell lines (iPSC)
No statistical methods were used to estimate sample size. iPSCs obtained from healthy donors of European descent were selected from the HipSci Consortium[12]. None of cell lines were reported to be commonly misidentified according to ICLAC v13. All HipSci samples were collected from consenting research volunteers recruited from the NIHR Cambridge BioResource (bioresource.nihr.ac.uk/studies/cbr62/; NIHR BioResouce Study Code: CBR62). HipSci samples were originally collected under ethical approval for induced pluripotent stem cell derivation (REC 09/H0304/77, V2 04/01/2013). Subsequent samples were collected following an updated consent (REC 09/H0304/77, V3 15/03/2013).

### Differentiation of iPSC into macrophages
HipSci cell lines were differentiated to macrophages based on a previously published protocol[67] which is described in detail in Supplementary Note 1. Of 315 lines initially selected, 227 (71.6%) were successfully differentiated. RNA-seq libraries were produced for 217 iPSC cell lines and 209 lines remained after quality control. Differentiated macrophages have been shown to be transcriptionally similar to monocyte-derived macrophages[13]. The differentiated naive macrophages were then stimulated with a diverse panel of adjuvants, resulting in 20 stimulation conditions, as well as naive differentiated (Ctrl_6 and Ctrl_24) and undifferentiated controls (Prec_D0 and Prec_D2). mRNA was harvested 6 and 24 hours after stimulation. Multiple cell lines were derived from a varying number of individuals in each stimulation condition (ranging between 177-202 individuals; Supplementary Data 9), resulting in a total of 4698 unique RNA-seq libraries across all conditions.

### Genotype imputation and quality control
Individuals who donated cell lines were previously genotyped through the HipSci project[12]. Genotypes were obtained for each cell line from the HipSci Consortium (hipsci.org) and genotype calling and imputation to UK10K and 1000 Genomes Phase I is described in ref. 11. We used CrossMap[68] to lift over from GRCh37 to GRCh38. Imputed variants with a low imputation score (INFO < 0.4), Hardy-Weinberg

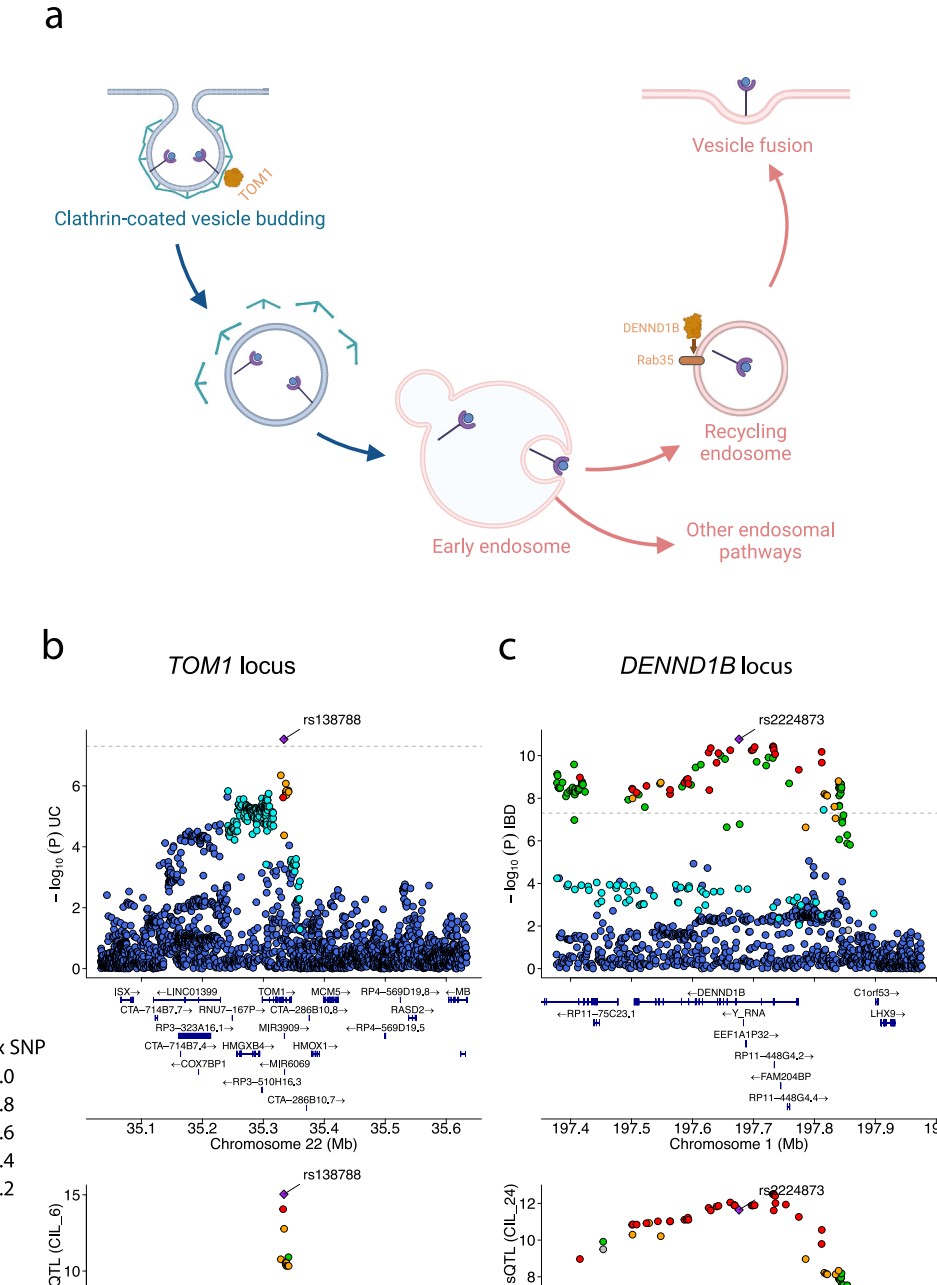

**Fig. 7 | *TOM1* and *DENND1B* sQTLs colocalise with Crohn's disease signals.**
**a** Diagram showing different stages of vesicle-mediated transport. *TOM1* and *DENND1B*, two regulators of endosomal trafficking are shown (created with BioRender.com). **b** GWAS regional association plots for the UC locus at 22q12.3 and the *TOM1* sQTL in stimulated macrophages (in CIL_6) showing high colocalisation

evidence. **c** Regional association plots for the IBD locus at 1q31.1 and *DENND1B* sQTL in stimulated macrophages (CIL_24). Horizontal dotted lines indicate genome-wide significance (*P*-value = 5 × 10⁻⁸). Lead IBD and UC SNPs are indicated with an arrow. Colours represent linkage disequllibrium with the index SNP. "Created in BioRender. El Garwany, O. (2025) https://BioRender.com/d8jdwgf ".

equilibrium *P*-value < 10⁻⁶, a minor allele frequency (MAF) < 0.05 or a missingness rate > 0.001 were filtered out. For the remaining variants, genotype principal components (PCs) were calculated using EIGENSTRAT[69] to correct for population stratification.

### RNA-seq and quality control

To process the large number of libraries more efficiently, two RNA-seq library construction protocols were utilised, including a modified Smart-seq2 protocol and the NEBnext Ultra II Directional RNA Library kit (further details provided in Supplementary Note 1). RNA-seq reads (75 bp paired-end) were aligned to the GRCh38 reference human genome and GENCODE v27 transcript annotation using STAR v2.5.3a[70]. To quantify gene expression we used featureCounts v1.5.3[71]. We kept protein-coding and lincRNA genes in all QC analyses as well as genes with mean expression ≥ 0.5 transcripts per million (TPMs) in at least half of the conditions (≥12), resulting in a total of 14,060 genes. To ensure the quality of the samples, we employed several QC metrics. Principal Components Analysis (PCA) was performed per 96-library pool (4 iPSC lines per pool, 24 conditions per iPSC line) to detect sequencing outliers. Non-stimulated or mislabeled stimulated samples were identified and discarded based on pairwise PCA comparisons of each condition with the rest of the conditions, per 96-library pool. Sex incompatibility checks were also performed using the methods described in ref. 72 and 3 iPSC-lines (72 samples) were discarded due to discordant sex annotations. Subsequently, we performed UMAP analysis[73] to cluster the different conditions and wrongly labelled samples that passed PCA filtering were discarded. Fnally, we utilised the Match BAM to VCF (MBV) method[74] to detect sample swaps and cross contamination between RNA-seq samples. We discarded 3 iPSC-lines (72 samples) and 63 additional samples due to cross contamination, and corrected the labels for 23 iPSC-lines identified as swaps. We did not observe concordance of genotype-RNA-seq data for 4 lines which we kept in the final dataset for differential splicing analysis but discarded from sQTL mapping. Among the 23 swaps, 2 lines were identical with lines already present in the data and were subsequently removed from the dataset.

In total, we discarded 8 iPSC-lines (-3.7% of the successfully differentiated lines) and 510 RNA-seq samples (-9.8%) based on our QC metrics (318 samples based on all QC metrics, 192 from the discarded 8 iPSC-lines) resulting in a total of 4698 unique RNA-seq libraries across all conditions.

### RNA-seq mapping for splice junction quantification

For the purpose of quantifiying splice junction usage, we remapped RNA-seq data using different parameters from the ones used for the initial QC in the previous section. Additionally, we wished to remove split reads with splice junctions overlapping genetic variants whose alleles result in ambiguous mapping quality (see section WASP filtering of ambiguously-mapped reads). BAM files mapped in the initial RNA-seq QC step were reverted to FASTQ files using the samtools v1.12[75] using the command: samtools collate -u -O BAM| samtools fastq −1 FASTQ.1 −2 FASTQ.2. FASTQ files were then mapped to the reference genome build GRCh38 using splice-aware aligner STAR v2.6.1[70] using the following parameters:

"--twopassMode Basic --outSAMstrandField intronMotif --out-SAMtype BAM SortedByCoordinate --outSAMunmapped Within --outSAMattributes All --outFilterMismatchNoverReadLmax 0.04 --outSAMmultNmax 1 --limitBAMsortRAM 40000000000 --sjdbOverhang 74 --waspOutputMode SAMtag"

This step outputs BAM files that were then used for the identification of split reads using LeafCutter.

### WASP filtering of ambiguously-mapped reads

It has been shown that ambiguously-mapped reads (RNA-seq reads that map to multiple genomic positions) can bias the number of split reads mapped across splice junctions[76].

WASP identifies an ambiguously-mapped read by replacing variant positions with the alternative alleles and then remapping[76]. In the previous step, we used STAR with the "waspOutputMode" which flags reads that pass the wasp filter with "[vW]=1". We therefore only removed read alignments which did not have tag "[-vW]=1" using this command in SAMtools[67]:

"samtools view -S -b -e "[vW]!=2 & & [vW]!=3 & & [vW]!=4 & & [vW]!=5 & & [vW]!=6 & & [vW]!=7"

### Identification of split reads

Split reads were identified from BAM files from the previous step using regtools[77] and output as.junc files. The following command and parameters were used:

"regtools junctions extract -s 0 -a 8 -m 50 -M 500000"

These parameters specify the minimum and maximum intron length (50 bp and 500,000 bps in length, respectively), and a minimum splice junction anchor length of 8 bps. The last parameter means that there must be reads supporting at least 8 basepairs of either sides of a splice junction.

### Intron clustering

Mapped split reads (as.junc files) were used to perform intron clustering using the leafcutter_cluster.py script.

python leafcutter_cluster_regtools_py3.py -j {input} -m 50 -l 500000

A minimum number of 50 split reads across samples was required to support an intron cluster. Maximum intron length used was 500 kbp.

For each identified intron cluster with introns 1...j, LeafCutter quantifies intron usage ratio R for intron k as:

$$R_k = \frac{X_k}{\sum_{i=1}^{j} X_i} \tag{1}$$

Where $X_i$ is the number of split reads that belong to an identified intron.

### Differential splicing analysis

Differential splicing analysis was performed between each of the 10 stimulated conditions and their corresponding timepoints controls (e.g. sLPS_6 vs Ctrl_6 and sLPS_24 vs Ctrl_24). LeafCutter differential splicing analysis tool (script leafcutter_ds.R)[12] was used with eight experimental covariates run ID, donor, library preparation method, sex, differentiation media,purity, estimated cell diameter, and differentiation time (Supplementary Data 2), and with the following parameters:

"−min-samples-per-intron 50 −min-samples-per-group 50 −min-coverage 30 −timeout 900"

### UMAP visualisation

UMAP was performed on raw intron usage ratios that were observed across all 24 conditions. UMAP clustering[73] was performed after 3 pre-processing steps. First, we performed quantile normalisation on raw intron usage rations. Second, we performed rank-based inverse normal transformation. Third, we regressed out eight technical covariates similar to the differential splicing analysis: run ID, Donor, library preparation method, sex, differentiation media, purity, estimated cell diameter, and differentiation time. We used the umap R package to perform UMAP clustering (github.com/tkonopka/umap).

## Intron usage ratio quality control and normalization

In order to use intron usage as quantitative traits, we first used the LeafCutter script "prepare_phenotype_table.py", which applies two default filters at the intron-level. It removes introns which have an intron usage ratio of 0 in more than 40% of samples, and those with a standard deviation of less than $5 \times 10^{-3}$. The script also applies quantile normalization to the introns which makes different samples have the same distribution of intron usage ratios. These introns were mapped to known intron-exon junctions in genes using the GENCODE v27 annotation in order to map its corresponding gene.

## Mapping sQTLs using intron usage ratios

sQTLs were mapped using normalised intron usage ratios and genotype data from samples within each stimulation condition. Variants in a 1 mega base pair (Mb) window around the transcription start site (TSS) were associated with the intron usage ratios. Genotype-intron association was modelled using a linear regression model implemented in QTLtools v1.3.1[78] with the parameters:

"-seed 1354145 --permute 1000 --window 1000000 --grp-best"

The option –grep-best allows phenotypes (intron usage ratios) to be organized in groups. Within each phenotype group, the genotype-phenotype sample labels are permuted in exactly the same way. This allows *P*-values for phenotypes within the same group to be compared with each other and for QTLtools to report the best associated phenotype per group. We group introns by the gene they belong to, and report the best associated variant-intron per gene.

In all of our analyses, we included the first three genotypic principal components (PC) as covariates. In order to remove unwanted variability in intron usage ratios, we mapped sQTLs separately in different conditions using 0, 1, 2, 3, 4, 5, 6, 7, 8, 9, 10, 20 and 50 intron usage ratio PCs as additional covariates to map sQTLs. We then counted the number of genes with significant sQTL effects (sGene) at a false discovery rate (FDR) $\leq 0.05$ using the R package qvalue v2.16.0[79]. In all our downstream analyses, we use the number of intron usage ratio PCs that maximises the number of sGenes discovered.

Three genotype PCs in addition to 0, 1, 2, 3, 4, 5, 6, 7, 8, 9, 10, 20, and 50 intron usage ratio PCs were used as covariates. In the rest of our downstream analysis, we map sQTLs with the number of PCs that maximise the number of genes with a significant sQTL effect (sGene; Supplementary Fig. 28). We call significant sQTLs at a false discovery rate (FDR) $\leq 0.05$ using the R package qvalue v2.16.0[79].

## Genome-wide summary statistics

GWAS summary statistics were downloaded from either the GWAS catalogue[34] or from UK BioBank GWAS summary statistics[80] (Supplementary Data 4). Summary statistics were formatted using a custom script so that each file contains at least: the chromosome and position (GRCh38) of associated variants, effect sizes, standard errors.

## Identification of genome-wide significant loci from IMD GWAS summary statistics and colocalisation analysis

To identify genome-wide significant loci, we identified all variants that passed a *P*-value threshold $< 5 \times 10^{-8}$. We ordered these SNPs from lowest to highest *P*-value and then iterated over each of them. At each iteration, we define a 1 Mbp window centred around the SNP. If a SNP falls within a locus that has been defined in a previous iteration, it is not used to define a new locus.

For each genome-wide significant locus, we first identified the set of all genes whose TSS falls within the locus. We performed pairwise colocalisation tests for all the intron usage ratios (within each of these genes) and the GWAS association signal. We perform these tests within each of the 24 conditions.

We matched variants between the sQTL and GWAS association signals by position on GRCh38 only since not all GWAS studies provided alleles. We perform colocalisation analysis using the R package coloc v5.1.1[32] and we discard GWAS-intron pairs that have <20 variants in common. Only SNPs in common between each GWAS study and our genotypes are considered. This outputs a list of tested introns per GWAS per condition. This exact same procedure was used for GTEx sQTLs.

Locus plots for the colocalisation were produced using the locuszoomr R package[81]. Linkage disequillibrium values were generated from Non-Finnish Europeans from the 1000GP high coverage project[82]. Heatmaps for visualising PP4 values were generated using the ComplexHeatmap R packages[83].

## Condition-specificity analysis using mash

We tested the condition-specificity of sQTLs using mashR v0.2.57[27]. mashR is an adaptive shrinkage framework that can be used to compare effect size estimates in a multi-tissue or multi-condition association study. It outputs shrunk effect size estimates in addition to a statistic indicating if effect sizes between two conditions are significantly different from each other (local false sign rate; LFSR). We use this framework to test if a given sQTL effect is a "response sQTL", meaning that its effect size is significantly different in a stimulated macrophage condition from unstimulated macrophages.

mashR requires training data to learn the correlation structure in the data from a set of canonical and data-driven covariance matrices (i.e. adaptive). The data-driven matrices are usually derived from the strongest QTL associations in the data. mashR then uses a randomly sampled set of QTL association to learn the mixture components of the provided covariance matrices. We use the lead sQTL SNP per gene (for genes with an FDR $\leq 0.05$) as our strong subset, and $10^6$ randomly sampled sQTL associations as the random subset (effect sizes and standard errors). Mash can also be trained using a baseline condition, against which other conditions are compared (we use Ctrl_6 and Ctrl_24 for conditions where RNA was harvested 6 and 24 h following stimulation, respectively).

The learned model can be used to recalculate "posterior" summary statistics for any desired set of sQTL associations by providing their effect sizes and standard errors. In our analysis, we recomputed effect sizes and LFSR for two sets of sQTL effects:

1-Lead SNP for significant sQTLs (FDR $\leq 0.05$) per sGene within each stimulated condition (Fig. 3b). This was used to estimate the number of sGenes with at least one response sQTL.

2-Lead SNP for colocalised genes (PP4 $\geq 0.75$) in the condition in which the colocalisation was detected. This was used to estimate the proportion of colocalised genes with a response sQTL within each stimulated condition (Fig. 4c).

## Colocalisation between significant sQTLs and eQTLs

We performed statistical colocalisation between all significant sQTLs (FDR $\leq 0.05$) to investigate whether the sQTL and eQTL signals in our sGenes are likely to be driven by the same causal variant. We only colocalised the intron with the highest *P*-value per sGene.

## RNA-seq coverage plots

RNA-seq coverage plots were generated using custom scripts based on pyGenomeTracks[84] with an additional sashimiBigWig track from Mu et al.[49,85]. Wrapper scripts are available at github.com/andersonlab/macromapsqtl/tree/main/sashimi. Briefly, RNA-seq coverage and intron usage ratios were averaged within each genotype group for a given variant using the commands samtools view and samtools merge. RNA-seq coverage was then calculated using the command bamCoverage using a binsize of 10 and output as a bedgraph file. Bedgraph files were then converted to bigWig files using the command bedGraphToBigWig[86] which were then visualised using pyGenomeTracks.

## Tissue dissociation and single-cell preparation

15 terminal ileum (TI) biopsies from eight healthy and seven non-inflamed CD individuals were dissociated following a single-step digestion protocol we previously developed[39]. Terminal ileal biopsies were dissociated using cold enzymatic and mechanical methods designed to preserve the viability of epithelial, immune, and stromal cell populations. Biopsies were first finely minced and pipetted to release immune cells into solution, referred to as *fraction 1*. The remaining tissue fragments were transferred to calcium- and magnesium-free HBSS supplemented with 2 mM EDTA, 0.26 U/µl serine endoprotease from *Bacillus licheniformis* (Sigma, P5380), 5 µM QVD-OPh (Abcam, ab141421), and 50 µM Y-27632 dihydrochloride (Abcam, ab120129). This suspension was incubated on ice for 30 minutes with intermittent pipetting to facilitate the release of epithelial and stromal cells (*fraction 2*).

Cells from both fractions were pooled, washed, and centrifuged before undergoing a 10-minute digestion at room temperature in HBSS containing $Mg^{2+}$ and $Ca^{2+}$ (phenol red–free), supplemented with 5 mM $CaCl_2$, 1.5 U/µl collagenase IV (Worthington, LS004188), and 0.1 mg/ml DNase I (Stem Cell Technologies, 07900). The resulting cell suspension was passed through a 30 µm filter (CellTrics, 04-0042-2316), washed, centrifuged, and treated for 3 minutes with ACK lysis buffer (Gibco, A10492) to eliminate red blood cells. Two final wash/centrifugation steps were performed, followed by filtration through a 40 µm mesh and manual cell counting using a haemocytometer (NanoEnTek, DHC-N01).

## Terminal ileum biopsies processing with 10X

Single-cell RNA sequencing was undertaken using 3′ 10X Genomics kits (v3.1) according to the instructions of the manufacturer. We targeted 6000 cells for CD participants and 3000 cells for healthy participants to account for the increased cellular heterogeneity in CD biopsies. Number of detected cells per sample is provided in Supplementary Data 5.

## Long-read RNA-seq data generation

Full-length cDNA obtained from single cell RNA sequencing 3′ 10X Genomics kits was used to generate single cell RNA long read sequencing employing MAS-Seq library preparation method[40] for 10x single cell 3′ kit (102-407-900). MAS-seq library preparation kit enables the concatenation of up to 16 cDNA molecules into a longer cDNA molecule that is subsequently sequenced using consensus circular sequencing. This approach achieves high efficiency while maintaining high accuracy. 15-75 ng of single cell full length cDNA was used as input and library preparation protocol was followed according to the manufacturer's instructions. Sequencing was performed using the Revio long-read sequencing system where each sample was loaded on a single SMRT cell.

## Cell type annotation

A Celltypist[87] (v1.6.2) model trained on short-read single-cell RNA-sequencing terminal ileal data[78] was applied to the single-cell short-read data from the equivalent 15 samples' cells. Cells which could both be confidently annotated in the short-read data as an ileal cell type (Celltypist confidence > 0.5) and were not predicted to be multiplets (as determined by Scrublet v0.2.1[88]) were retained, and those that could not were discarded. The cell type and category labels from the annotated short-read data were transferred directly to the long-read data using matching sample identifiers and barcodes to identify the matching cells. Cells which did not have an equivalent barcode in the long-read or short-read data were discarded.

## Long-read RNA-seq data quality control and identification of transcripts

We obtained BAM files containing HiFi reads (>Q20) and performed quality control using the standard isoseq[89] pipeline by Pacbio.

Concatenated reads were bioinformatically deconcatenated using skera v1.2.0 with the default parameters. Remaining unwanted concatemers and polyA tails were removed with lima v2.9.0 with the default parameters. Cell barcodes were then identified and truncated as a separate BAM tag using isoseq tag (BAM tag CB).

In all the following steps, we used isoseq v4.0.0. We used the "isoseq correct" command to correct possible sequencing errors in barcodes. By default, isoseq correct removes any cell barcodes with edit distance > 2 from the 10 × 3′ barcode whitelist (http://downloads.pacbcloud.com/public/dataset/MAS-Seq/REF-10x_barcodes/3M-february-2018-REVERSE-COMPLEMENTED.txt.gz). Finally, to filter out cDNA generated by ambient RNA molecules, we quantified the number of UMIs per cell barcode. Percentiles of UMIs per cell barcode were calculated and cells with a UMI per cell count at the 99th percentile were kept (using the isoseq correct –method percentile --percentile 99). Finally, duplicate UMIs were deduplicated using the isoseq command isoseq groupdedup with default parameters.

Deduplicated reads from real cells were then mapped to the human genome hg38 using pbmm2 align –preset ISOSEQ, which is based on minimap2. This preset uses the following minimap2 options: -k 15 -w 5 -u -o 2 -O 32 -e 1 -E 0 -A 1 -B 2 -z 200 -Z 100 -r 200000 -g 2000 -C 5 -G 200000. Briefly, these options specify the match/mismatch penalties (-A and -B), -k specifies the k-mer size and -w specifies the minimizer window. -z and -Z specify the Z-drop and Z-drop inversion scores which are used by minimap2 as part of the Z-score heuristic to eliminate alignments with sudden drops in alignment quality along the diagonal elements of the dynamic programming matrix. -C is the cost of non-canonical splice junction (GT-AG) and -G is the maximum intron length. -e, -E, -o and -O specific the two gap extension and the two gap opening penalties (the cost of a k-long gap is min{o + k*e,O + k*E}).

After mapping to the genome, we used the mapped BAM files as input to the isoseq collapse command which merges transcripts that have the same exon and intron chains. Additionally, isoseq collapse merges isoforms which are truncated at the 5′ end with longer transcripts[90]. It produces a GFF of non-redundant isoforms and an accompanying abundance file.

Using the GFF file from isoseq collapse, we used SQANTI3[91] to perform quality control and filter out isoforms resulting from potential library preparation artefacts. However, it is worth noting that for the purpose of validating low-usage splice junctions, we are not necessarily interested if a splice junction comes from a complete or truncated isoform as long as the identified isoform splice junctions do not represent a library preparation artefact. To this end, SQANTI3 identifies artefacts that may lead to erroneous splice junction identification (using the sqanti3_qc.py and sqanti3_filter.py scripts):

1- poly-A intrapriming artefacts as isoforms whose transcription termination site belongs to a genomic poly-A rich region. Specifically, if the 20 basepairs downstream are > 60% A bases, an isoform is classified as being a result of poly-A intrapriming (SQANTI3 rule: "perc_A_downstream_TTS":[0,59]).

2- SQANTI3 identifies reverse transcription switching artefacts as apparent splice junctions whose adjacent sequences are direct repeats of one another, which is a well-known characteristic of reverse transcription artefacts[92] (SQANTI3 rule: "RTS_stage":False).

3- We applied a number of additional filters depending on the relationship between the discovered isoform and the closest annotated isoform. For isoforms with a full or partial annotation match (full splice matches or FSM and incomplete splice match or ISM), we filtered out mono-exonic isoforms, reverse transcription switching artefacts and intra-priming artefacts. For all other types of isoforms (partial matches or ISM), novel-not-in-catalog (NNC) and novel-in-catalog (NIC), we applied an additional condition that all splice junctions are canonical (GT-AG; SQANTI3 rule "all_canonical": True).

After removing artefacts identified by SQANTI3, abundance files and GFF files were used as input to pigeon make_seurat to obtain a cell-by-isoform matrix (.mtx format). From the GFF file, intron chains were obtained using a custom script and a low-usage splice junction was considered validated if it was present in at least one isoform in the cell-by-isoform matrix.

## Reporting summary

Further information on research design is available in the Nature Portfolio Reporting Summary linked to this article.

## Data availability

Imputed genotype data for the HipSci lines are available from the European Nucleotide Archive (ENA) (PRJEB11749) and European Genome–phenome Archive (EGA) (EGAD00010000773). Unprocessed RNA-seq data generated in this study have been deposited in the EGA database under study accession code EGAS00001002268 (EGAD00001015380). Full sQTL summary statistics (both nominal and permutation passes), colocalisation results and intron usage ratios (raw pre-QC and normalised post-QC) are publicly available at ftp.sanger.ac.uk/pub/project/humgen/summary_statistics/macromap_sqtl/FTP/. Long-read cell-level counts, cell-type annotations, GTF files and SQANTI3 output files are available at https://ftp.sanger.ac.uk/pub/project/humgen/summary_statistics/macromap_sqtl/FTP/long_read_data/macromap_deposit/. We have made significant sQTL data (False Discovery Rate ≤ 0.05) publicly available via an interactive portal (www.macromapqtl.org.uk/sqtl/).

## Code availability

The code used is available at github.com/andersonlab/macromapsqtl and zenodo.org/records/15156885 (https://doi.org/10.5281/zenodo.15156885).

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

## Acknowledgements

We would like to thank the Sanger Institute Scientific Operations teams and Human Genetics Informatics team for providing sample handling, data generation and computational support to enable the analyses described in this manuscript. This work was supported by Wellcome Sanger Institute Core funding from the Wellcome Trust (206194, 220540/Z/20/A, 108413/A/15/D). The iPSC lines were generated at the Wellcome Sanger Institute, under the Human Induced Pluripotent Stem Cell Initiative funded by a strategic award (WT098503) from the Wellcome Trust and Medical Research Council. Collection of gut biopsies was supported by the NIHR Cambridge Biomedical Research Centre (BRC-1215-20014). The views expressed are those of the authors and not necessarily those of the NIHR or the Department of Health and Social Care. N.I.P. was supported by the Early Postdoc Mobility fellowship from the Swiss National Science Foundation (178005). M.I. supported by core funding from the British Heart Foundation (RG/18/13/33946), NIHR Cambridge Biomedical Research Centre (IS-BRC-1215-20014), BHF Chair Award (CH/12/2/29428) and Cambridge BHF Centre of Research Excellence (RE/18/1/34212). We would like to thank Emma Davenport for reviewing this manuscript and providing valuable feedback.

## Author contributions

O.E.G. performed the analyses and, along with N.I.P., C.A.A. and D.J.G. drafted the manuscript with contributions from all authors. N.I.P. assisted with multiple analyses and independently validated sQTL results and provided feedback to improve the manuscript. N.K. provided statistical feedback on several occasions, assisted with multiple analyses, and conducted analysis on the pilot data. L.B.V., A.T. (Anthi Tsingene) and C.G. applied the differentiation protocol, performed QC metrics on the differentiated macrophages and carried out the stimulations under the supervision of C.G. A.K. optimised the low-input bulk RNA-seq protocol and prepared the RNA-seq libraries along with M.I. and A.B. C.C.M. generated the long-read sequencing libraries from terminal ileum biopsies provided by T.R. M.K. and T.A. performed the cell type annotation that was used to match cell barcodes between short-read and long-read cell barcodes. D.J.G. conceptualised the study while C.A.A. and D.J.G. supervised this work. A.T. (Alex Tokolyi) built the interactive web portal that hosts the summary statistics data.

## Competing interests

C.A.A. has received research grants or consultancy/speaker fees from Genomics plc, BridgeBio, G.S.K. and AstraZeneca. D.J.G. was an employee of BioMarin and N.I.P. was an employee of G.S.K. at the time the manuscript was submitted. The remaining authors do not declare any competing financial or non-financial interests.
