## [Transparent Peer Review file · Nature Communications]

Splicing QTL mapping in stimulated macrophages associates low-usage splice junctions with immune-mediated disease risk

Corresponding Author: Dr Carl Anderson

Version 0:

Reviewer comments:

Reviewer #1

(Remarks to the Author)

This study aimed to assess the contribution of alternative splicing to the risk of IMD by mapping sQTL variants in macrophages exposed to various cellular conditions. The sQTL analysis was conducted using standard protocols such as STAR for mapping, LeafCutter for intron clustering, QTLtools for sQTL mapping, and so on. However, many of the findings presented may have already been reported in previous sQTL studies. Therefore, the authors may need to provide further clarification on the scientific advancements their findings have brought about.

Major points.

1) Because the previous study conducted by Rotival et al. (ref 53) also performed sQTL analysis on macrophages stimulated with various stimuli, their paper should be referenced appropriately in the introduction. Furthermore, the authors should elucidate the novel findings in the current study when compared to those presented in Rotival's paper.

2) The authors mentioned that eQTL analysis has also been conducted using the same dataset employed in the present study (line 211). If this is the case, the authors should explicitly state that the present study relied on the secondary use of existing data, which could potentially diminish the value of this study as a data resource.

3) Since this study evaluated iPS-derived macrophages rather than primary macrophages, it could have significantly influenced the analysis results. Although the authors mentioned that "macrophages differentiated using our experimental protocol have been shown to be transcriptionally similar to monocyte-derived macrophages (line 477)," it would be beneficial if the authors could further investigate potential differences in the expression of major spliceosome genes between primary and iPS-derived macrophages.

4) Is there any evidence, such as histone marks or CAGE-peaks, indicating the presence of alternative transcription start sites (TSS) for PTPN2-205? The case of PTPN2-205 may raise a question regarding the author's claim that "unlike eQTL variants, sQTL variants tend to be located around intron splice sites rather than the TSS of genes (Supplementary Figure S3), closer to intronic GWAS association signals that do not colocalize with eQTL signals." The sQTL variant at this locus (rs80262450) is located not only around the intron splice site but also around alternative TSS.

5) The major claim of this study (as indicated by the title) is that low-usage splice junctions underlie the risk of immune-mediated diseases. However, only the percentage of UIR < 0.1 sQTL in the entire dataset (53.4%) was presented. Could this proportion be affected by stimulation?

6) The findings related to PTPN2-205 are crucial for supporting the author's claim regarding the significance of low-usage splice junctions. However, I have several concerns regarding the interpretation of these results. Firstly, the sQTL SNP rs80262450 exhibits a strong eQTL signal for whole blood cells in the GTEx data (it also has a strong eQTL effect for RP11-973H7.1 in colon tissues). Therefore, the absence of colocalization observed for eQTL in the macrophages under 13 different conditions does not necessarily rule out the possibility that eQTL is the true causal factor at this locus. Furthermore, the authors speculated that increased usage of PTPN2-205 might attenuate the role of PTPN2 as a negative regulator (line 333). However, given that this transcript consists of only three exons, it raises a question about how it could function as a

regulator. To address this issue, experimental validation may be necessary.

7) The GENCODE annotation used in this study might be too old (v27) to precisely evaluate the "low-dosage" junctions. Would the findings substantially change if the most recent annotation (v44) were used? In particular, the author's claim that "Moreover, 50.6% of these introns are not found in any annotated transcripts in GENCODE v27, whereas only 12% of introns with a mean IUR ≥ 0.1 are absent from GENCODE v27 (Figure 6b)" should be based on the recent annotation.

Minor points.

8) Please show the location and structure of genes in Figure 7b and c.

9) I could not find rs80626450 (line 316) in dbSNP.

Reviewer #2

(Remarks to the Author)

This manuscript examines splicing quantitative trait loci in macrophages exposed to various cellular conditions. They report a large number of genes harboring sQTL effects, many of which showed colocalization with immune mediated disease-associated risk loci. They observed that about half of the colocalizations involved lowly-used splice junctions. An example event in PTPN2 was shown. The findings support the potential role of alternative splicing in immune mediated disease (IMD) risks. However, there are some major concerns that need to be addressed.

In comparing sQTL prevalence across conditions (Fig. 3c), power is a major issue. The authors should do the comparison in the context of a power evaluation. The events are comparable only if they have similar power in different conditions.

A major concern is with the main finding claimed by the study that many lowly observed splicing events may underlie IMD risks. It is well known that RNA-seq analysis may yield spurious spliced junctions, which may lead to identification of low usage splicing junctions that are indeed false positives. The authors need to carry out rigorous quality checks in read mapping and splicing event identification to rule out this possibility. Since this is a major conclusion of the paper (in the title), this issue needs to be investigated very thoroughly.

For sQTL introns with a mean intron usage ratio (IUR) < 0.1 , what are the IUR values for each genotype? What are the effect sizes? What are the power in detecting them as significant sQTL? These data should be shown in detail. For each intron, are all genotypes associated with low IUR, or certain genotypes with relatively high IUR (e.g. > 0.1) and others with low values? If it is the latter case, then it does not make much sense to call them lowly used.

Experimental validations are needed to confirm the existence of true lowly used splicing isoforms, as this is a major conclusion of this paper.

In the PTPN2 example, it is not clear if the 205 isoform is actually expressed because Fig. 6d does not seem to show read distribution overall the first exon of this isoform which is unique to 205. This needs to be clarified and experimental validation should be conducted to confirm its expression.

The PTPN2 example also brought up a question about effect size of the sQTLs. It seems the different genotypes have very small differences in their IUR. With such a small splicing change, it is not reasonable to assume a functional impact exists. This undermines the premise for the entire paper. The authors should examine effect sizes of sQTLs, and apply a requirement on minimum effect sizes that make biological sense.

Page 14, line 376-378, the conflicting directions of effect in different conditions may be interpreted as weak and unreproducible results, rather than context-dependent results. It is necessary to consider this and prove they are not false positives.

Minor:

Figure 2a, b: consider providing more detailed labels to show both conditions and timepoints, it is currently very hard to read.

The variants with colocalization signals are not necessarily causal. Please avoid using such a strong term.

The Results section contained extensive discussions. Suggest moving those to Discussion.

Reviewer #3

(Remarks to the Author)

This manuscript studies how genetic variants are associated with alternative splicing in macrophages derived from iPSCs. They focus on how splicing QTLs may be contributing to immunological diseases by using GWAS data. The authors developed a macrophage activation map where they stimulated macrophages with ten different stimuli at two different time points. They demonstrate how alternative splicing varies among specific stimuli, particularly in innate immune response genes. They observed that sQTLs that co-localize with GWAS loci often involve low abundance introns. They provide

specific examples of their findings. For instance, they discovered an sQTL for the PTPN2 gene that is specific for activation conditions and that implicates a low abundance intron. This sQTL colocalizes with an IBD risk locus. Additionally, they identified two other colocalization signals for LRRK2 and DENND1B in IBD, which participate in the same pathway. In summary, the authors report a novel macrophage sQTL map, aiding in the understanding of immune-mediated diseases and highlighting the importance of splicing and cell state context.

The manuscript is well written and well structured. The data used in this manuscript were previously generated and constitute a very large collection of useful data for the field. A focus on the splicing aspect of this dataset is insightful and worthy of publication. However, several of the analyses feel a bit “rushed” and I believe deeper and more thorough analyses are necessary to support the main claims of the paper.

Major concerns

1. Replication of macrophage sQTLs. In most QTL studies people check if their QTLs replicate compared to other QTL studies in the same cell-type. Given these are iPSCs derived macrophages, I believe replication in this study would be of particular relevance. It is important to know how similar the QTLs found in this study are to primary macrophages. There are multiple macrophage QTL studies in eQTL Catalogue from which summary statistics could be easily used, given they also used LeafCutter for splicing quantification.

2. Colocalization analyses.

a. The authors use different PP4 thresholds depending on their analyses, with little information on the rationale behind. Especially for considering an sQTL-only colocalization with GWAS, they require sQTL PP4 > 0.75 and eQTL PP4 just needs to be lower than 0.75. I imagine there could be a lot of cases where eQTL PP4 is in the range of 0.65-0.74, and in that case they are not considered to colocalize with GWAS, when in an earlier analysis they used PP4 > 0.1 as a threshold of colocalization. The current threshold could be over-estimating how many cases are sQTL-only loci. Have the authors explored how the results change at other thresholds or in a more continuous manner?

b. Given the authors are claiming that sQTLs play such an important role in complex traits, have the authors considered performing colocalization analysis with more stringent methods, such as JLIM (Chun et al. Nature Genetics, 2017)? In the 21 IMDs selected, the authors found that close to 75% of risk loci are explained by co-localization with e/sQTLs of their macromap, while several of these top IMDs are known to likely have a different main cell-type mediating disease risk. For example, in studies assessing enrichment of heritability in cell-type specific annotations, PBC appears more of a B cell driven disease, MS is considered a T cell and B cell driven disease, and for childhood-onset asthma they have reported a T cell component. Can the authors comment on these results (which in my opinion are unexpected)? Perhaps the authors can include diseases that are known to have a strong macrophage/myeloid component (such as Alzheimer disease, Gjonneska et al., Nature, 2015), and some “negative control” diseases that are expected to not have an immune component, such as psychiatric diseases.

c. In the methods the authors specify that in their colocalization analyses they are including any locus that shares at least 2 SNPs between the GWAS and QTL study. This threshold seems quite lax, would the results change if the authors require a higher number of shared SNPs in the locus in order to test for co-localization with more information? (e.g. 20).

3. Depth of analyses for stimulation conditions.

a. It would seem only a couple of the stimuli were analysed in a focused way. For the pathway enrichment analysis, only the condition sLPS_6 and the PIC_6 were described in the results section. How do the results of pathway enrichment look for the other stimuli? Do you get the same pathways enriched across stimuli or do they change, and are the differences expected given the activation conditions?

b. The authors focus their analyses on comparing non-stim condition with 6hr or 24 hour activation, but little is shown regarding differences between activation conditions. Can the authors comment on activation condition-dependent sQTLs? How important is it to stimulate different pathways to capture disease relevant sQTLs or would one activation condition capture most of the disease associated effects?

c. I would also like to see in the text more details about the stimuli applied to the macrophages, and the rationale behind. A bit of that is shown in the scheme of Figure 1, but nothing is mentioned in the main text, and it is an important component of the study.

4. Lowly-used alternative splicing events and disease risk.

a. The authors show in Fig. 5 the “distribution of mean intron usage ratio for colocalised introns, showing a peak close to 0”. How does the distribution of intron usage ratio look for introns in general? Is there really a bias towards low usage in colocalized sQTLs or is that how the expected distribution look like?

b. The authors chose to consider only one intron usage per sQTL gene, choosing the one with the lowest P-value. Can choosing only the intron with lowest P-value also bias the results? Have the authors tried to include sQTLs of other introns of the same gene? Would those “other” introns that colocalize with GWAS also introns of low usage? Or are introns with lowest P-value per gene biased towards those of lowest usage?

c. Are these lowly abundant alternative splicing events implicated in disease enriched in non-sense mediated decay, as it is shown in the paper: Fair et al, bioRxiv, 2023? How many of these lowly-used alternative splicing events are likely to be translated into protein? Can you validate experimentally some of them?

Minor concerns

- In general, it would be great to see more visual examples of how the sQTLs look, especially for response QTLs. For instance,

for lines 138-143, it would be great to have a figure showing this exon usage.

- Line 512: "RNA-seq libraries were produced for 217, which represented 209 lines after quality control". Is a word missing after "for 217".
- Line 275: typo (figure 5c not 6c)
- Line 278: typo (figure 5b not 6b)
- Supplementary Figure S5 is odd, PTPN2 is shown for eQTL but not for sQTL
- Figure 6d : the legend is not explaining what are the blue/purple and yellow colors (correspond to genotype of sLPS_6) and I would like to see the same plot for the control condition
- Would be good to see the same plot as in Fig 6d for DENND1B and LRRK2

Version 1:

Reviewer comments:

Reviewer #1

(Remarks to the Author)

I thank the authors for responding to my comments, which I think improved the manuscript. I have several additional comments regarding their responses.

1)"Rotival et al. have profiled the alternative splicing landscape of monocytes exposed to four different stimuli, but have not robustly linked sQTLs to disease-associated loci."

Rovital et al. have indeed linked sQTL to disease-associated loci. Although I agree that the number of disease-associated loci is increased in the present study, I still think the scientific advance shown by the present study may not exceed that brought by Rotival's study.

2)"Our manuscript is submitted jointly with the MacroMap eQTL manuscript to the same journal(Panousis et al. 2023)"

I think this paper would be a split publication, but I would defer this issue to the journal editors.

(Remarks on code availability)

Reviewer #2

(Remarks to the Author)

Related to my original concern #2 and #5:

Because the major finding of this paper is related to low-usage splicing junctions, it is absolutely necessary to carry out experimental validations of the existence of such splicing events. This reviewer is not asking the authors to test the hundreds of hypotheses that your paper has generated (as you incorrectly interpreted in your response). At the minimum, the authors needs to experimentally test a random subset of the low usage splicing junctions, in order to confirm their existence and rule out the possibility of false positives.

Original concern #3: regarding effect sizes,

The figure provided is not showing what its legend says (specifically, y axis: absolute value of IUR). Based on the figure, it appears that low-usage (black) splice junctions had smaller absolute effect sizes than common-usage (orange) junctions. The reviewer understands the argument made by the authors about the normalization procedure. However, the effect size comparison between low usage and common usage junctions is important. The authors needs to examine this difference, and conclude whether low usage junctions are associated with smaller effect sizes relatively. Such data and analysis should be included in the manuscript.

Original concern #4: "For each intron, are all genotypes associated with low IUR, or certain genotypes with relatively high IUR (e.g. >0.1) and others with low values? If it is the latter case, then it does not make much sense to call them lowly used." The authors did not understand or interpret my question correctly. In their response, the authors state "A mean intron usage ratio < 0.1 indicates that across all individuals the mean intron usage is less than 0.1. As this is a mean across all individuals in the study, we believe it is appropriate to refer to them as low usage intron rations." This definition of "low-usage" splice junctions may be misleading because it is only based on a mean value. To be identified as significant sQTL events, the so called "low-usage" splice junctions should have genotypes that are NOT "low-usage" as sQTL looks for "a significant difference in mean intron usage with respect to genotype" (stated by the authors). Thus, calling these events "low-usage" based on the mean is misleading, which introduces an artificial sense of novelty to the manuscript.

(Remarks on code availability)

Reviewer #3

(Remarks to the Author)

The authors have substantially improved their manuscript and they have address my comments. My only remaining minor requests are the following:

- I hope the authors have given more experimental methods details of the stimulation conditions (including concentration of each stimulant used) in the accompanying Panousis et al. manuscript, but if not, please include those details in the methods section of this manuscript.

- The sQTL visualization plots that the authors have generated (such as in Fig. 6d) are really helpful to appreciate the events behind the sQTL co-localizing with disease risk loci. If the authors could share their code for creating those plots in their GitHub page, it would be greatly appreciated by the sQTL community (and it could increase their number of citations), especially given the LeafViz app does not allow visualizing splicing effects for more than 2 groups.

(Remarks on code availability)

I went into their github and saw their reported scripts. I didn't go deep enough to know whether the info that they provide is enough for replicating their main results. However, as I mentioned to the authors, if they make the code for visualizing sQTL available I believe it would be widely used.

Version 2:

Reviewer comments:

Reviewer #1

(Remarks to the Author)

No additional comment from this reviewer.

Reviewer #2

(Remarks to the Author)

I thank the authors for addressing my concerns.

Regarding the original concern #4: I understand the authors' response, and it addressed my question. However, to enhance scientific rigor, the authors should re-define "low usage" splice junctions. Instead of defining them based on a mean value, such junctions should be defined requiring all genotypes had IUR<0.1. This is important because it is not correct to call splice junctions "low usage" if they are not lowly used in certain people. The fact that such junctions had varying IUR levels across genotypes, where some IUR>0.1, makes it interesting.

Reviewer #3

(Remarks to the Author)

The authors have addressed my remaining comments. Thank you.

Response to referees

Manuscript: NCOMMS-23-33917A

Reviewer #1 (Remarks to the Author):

This study aimed to assess the contribution of alternative splicing to the risk of IMD by
mapping sQTL variants in macrophages exposed to various cellular conditions. The sQTL
analysis was conducted using standard protocols such as STAR for mapping, LeafCutter for
intron clustering, QTLtools for sQTL mapping, and so on. However, many of the findings
presented may have already been reported in previous sQTL studies. Therefore, the authors
may need to provide further clarification on the scientific advancements their findings have
brought about.

Major points.

**1) Because the previous study conducted by Rotival et al. (ref 53)**
**also performed sQTL analysis on macrophages stimulated with**
**various stimuli, their paper should be referenced appropriately in**
**the introduction. Furthermore, the authors should elucidate the**
**novel findings in the current study when compared to those**
**presented in Rotival's paper.**

We thank the reviewer for their suggestion. Rotival et al. investigated genetic and
evolutionary drivers of alternative splicing in monocytes obtained from 200 individuals of
African and European descent. They stimulated monocytes with LPS, P3C, R848 and
Influenza A virus. Their work demonstrated that differential isoform usage has been an
important substrate of innovation in the long-term evolution of immune responses and a
more recent vehicle of population local adaptation. In the discussion section of their
manuscript the authors state that “Future work using colocalization analyses and Mendelian
randomisation approaches should help establish a causal role of AS in disease risk at the
loci identified”. In our MacroMap sQTL work we take on this mission and plug this gap.

Other key differences between the studies and their findings include:

- 1. *The cell types in which sQTL were mapped.* Our study maps sQTLs in iPSC-derived
macrophages (and their precursors), while Rotival maps eQTLs in monocytes.
Macrophages and monocytes are different cell types and thus we expect the genetic
regulation of gene-expression to differ between them, especially after stimulation.
- 2. *The number of sQTL identified.* Our study identifies 5,734 sGenes in stimulated and
unstimulated macrophages (and their precursors), while Rotival identify 993 sGenes
in their stimulated and unstimulated monocytes. Thus, the results from our study
represent a uniquely rich resource of sQTLs for identifying effector genes for
diseases where macrophages are a relevant cell-type.

- 3. *The extent of insights into disease.* Unlike Rotival et al., our study had an explicit
focus on identifying sQTLs that are highly likely to underpin common complex
disease associations. We identified 707 unique genes with an sQTL signal that likely
shares a causal variant with a risk locus for at least one of 22 different immune-
mediated diseases ($PP4 \geq 0.75$; Figure 4a). In comparison, Rotival et al found only
27 sQTL that overlapped an immune system disorder GWAS locus. As such, the
results of our study provide a more impactful resource for identifying disease effector
genes and understanding the role that macrophages play in specific immune-
mediated diseases.
- 4. *The contribution of low-usage splice junctions to IMD risk.* We believe that we have
more strongly established a role for the dysregulation of low-usage splice junction in
IMD risk via colocalisation analysis. This finding opens the door for the functional
follow-up of these splicing events, and encourages a more comprehensive
characterisation of the transcriptome of immune cells.
- 5. *The extent to which sQTLs underpin eQTLs.* We undertake formal statistical tests of
colocalisation to explore the extent to which sQTL and eQTLs share causal variants.
We show that at most 25% of sGenes may share a causal variant with an eGene in
the same condition ($PP4 \geq 0.1$; Supplementary Figure 6; Methods). By comparison,
Rotival et al do not undertake formal colocalisation tests and instead they test the
effect of the peak sQTL SNP on intronic RPKM in the condition where the sQTL was
the most significant. This approach is more susceptible to confounding by linkage
disequilibrium than our colocalisation approach. Furthermore, Rotival et al found only
21/150 sQTLs where splicing modifications lead to gene expression changes (~14%
of tested sQTLs), whereas our estimate of 25% is based on 5,734 tested sGenes.
Taken together, we believe our upper bound for the extent to which sQTLs and
eQTLs share causal variants is more conclusive.

The Rotival et al manuscript is a good paper, which we have enjoyed reading. We don't see
it as competing with our paper. Indeed, we see that the Rotival et al. findings motivate our
study and thus we have changed the introduction of manuscript to better reflect this.

Changes to manuscript:

We have acknowledged both Nedelec. Et al and Rotival et al. in our Introduction and
summarised what our data and analyses add:

- • *“Previous work by Nedelec et al. has revealed ancestry-specific regulation of gene
expression in macrophages in response to Listeria and Salmonella. Moreover,
Rotival et al. mapped sQTLs in macrophages exposed to four bacterial stimuli and
found over 1,400 sQTLs. They highlighted an enrichment of disease-associated loci
among response sQTLs, but they have not systematically linked inherited changes in
alternative splicing to disease-associated loci. Similarly, Rotival et al. have profiled
the alternative splicing landscape of monocytes exposed to four different stimuli, but
have not robustly linked sQTLs to disease-associated loci.”*
- • *“Compared to previous work, our work expands on the number of environmental
conditions and takes on a disease-oriented approach to understand how disease-
associated loci impact alternative splicing in macrophages.”*

**2) The authors mentioned that eQTL analysis has also been**
**conducted using the same dataset employed in the present study**
**(line 211). If this is the case, the authors should explicitly state that**
**the present study relied on the secondary use of existing data,**
**which could potentially diminish the value of this study as a data**
**resource.**

Our manuscript is submitted jointly with the MacroMap eQTL manuscript to the same journal
(Panousis et al. 2023) and we hope that both get published together. The same raw data
underpins the two studies and will be made available upon publication of the two papers.
The sQTLs and the disease colocalisations we identify in this paper are themselves
important resources for the community that can fuel disease effector gene discovery and
biological understanding of a broad range of disease where macrophages play a role in
pathology. Therefore, we do not believe the co-publication of the gene-expression paper
diminishes the value of the sQTL and colocalisation resources from this paper.

***3) Since this study evaluated iPSC-derived macrophages rather than***
***primary macrophages, it could have significantly influenced the***
***analysis results. Although the authors mentioned that***
***"macrophages differentiated using our experimental protocol have***
***been shown to be transcriptionally similar to monocyte-derived***
***macrophages (line 477)," it would be beneficial if the authors could***
***further investigate potential differences in the expression of major***
***spliceosome genes between primary and iPSC-derived***
***macrophages.***

We would like to thank the reviewer for their suggestion. To address this, we identified
spliceosomal genes using the Reactome pathway database (splicing pathway; R-HSA-
72172). There were 193 spliceosomal genes in total, including both major and minor
spliceosomal genes.

In a previous paper, Alasoo et al., 2015 performed differential expression analysis between
primary macrophages and iPSC-derived macrophages using DEseq2. Importantly, the iPSC-
derived macrophages included in Alasoo et al. were derived following the same protocol we
used in the current paper. As such the results from the previously published analysis allow
120 us quantify the extent to which studying our iPSC-derived macrophages rather than primary
macrophages is likely to have impacted our current findings.

Of the 193 spliceosome genes, only 37 had were significantly differentially expressed
between primary macrophages and our iPSC-derived macrophages (adjusted P-value < 0.05).
However, the log-fold changes associated with these gene expression differences were
small. *SRSF2* had the largest fold change ($\log_2FC=0.79$) across the 193 genes, and thus we

think that differential expression of these 37 spliceosome genes is unlikely to bias the
splicing changes we observe in iPSC-derived macrophages.

Changes to manuscript

The following text was added the Results subsection "Macrophage stimulation initiates an
alternative splicing programme in innate immune response genes":

*"Major and minor spliceosomal genes were not differentially expressed between iPSC-
derived and monocyte derived macrophages (SRSF2 had the largest fold change
$\log_2FC=0.79$; Data from Alasoo et al. 2015)."*

**4) Is there any evidence, such as histone marks or CAGE-peaks,
indicating the presence of alternative transcription start sites (TSS)
for PTPN2-205? The case of PTPN2-205 may raise a question
regarding the author's claim that "unlike eQTL variants, sQTL
variants tend to be located around intron splice sites rather than
the TSS of genes (Supplementary Figure S3), closer to intronic
GWAS association signals that do not colocalize with eQTL
signals." The sQTL variant at this locus (rs80262450) is located not
only around the intron splice site but also around alternative TSS.**

The reason why we attributed the individual variation in splice junction usage to variation in
expression of transcript PTPN2-205 is that it was the only annotated transcript that contains
the implicated splice junction (chr18:12,817,365-chr18:12,818,944). As our dataset is
composed entirely of short reads, it is difficult to identify which transcript(s) those short reads
may have come from. Therefore, as stated in our manuscript, we could only hypothesize that
the sQTL is associated with the abundance of PTPN2-205, and confirmation of this is
required via long-read sequencing.

We did not find any FANTOM5 CAGE peaks mapping close to the TSS of PTPN2-205.
Notably, FANTOM5 included 101 RNA-seq samples obtained from monocyte-derived
macrophages in different stimulation conditions. None of these harboured CAGE peaks
within 1kbp of the first splice junction in PTPN2-205 (chr18:12,818,944). This prompted us to
investigate the possible identity of the transcripts that may have given rise to the colocalised
sQTL. One source of uncertainty is the incomplete nature of the GENCODE annotation, with
long-read RNA-seq studies showing that a large number of novel transcripts remain partially
or completely unannotated [2,3]. We therefore searched for transcripts that include the
colocalised PTPN2 splice junctions in long-read RNA-seq data.

 *Figure: Four PTPN2 transcripts feature the splice junction chr18:12817365-12818944 in a Pacbio PBMC sample*
 *sequenced using long-read RNA-seq (highlighted in green). All PTPN2 isoforms in GENCODE v45 are shown*
 *(highlighted in yellow).*

Long-read RNA-seq from two PBMC samples sequenced on the Pacbio Revio platform were
 recently made publicly available. These data have been used to identify a set of transcripts
 with supporting long reads. We found that four isoforms contain the implicated splice
 junction. The 3' splice site had an average depth of 1130 reads as it was an acceptor splice
 site in multiple *PTPN2* isoforms, many of which had high usage. The 5' site had an average
 depth of 6.5 reads. This coverage remained consistent even after excluding reads with
 significant intronic overlap (> 30% of the reads do not overlap with annotated exonic
 junctions). In only one of these four transcripts was the implicated splice junction the first
 splice junction in the transcript (Figure). It is therefore unclear whether the sQTL identified in
 our study represents an alternative splice site or an alternative transcription start site.

Given this uncertainty, we do not believe our findings at the *PTPN2* locus should raise a
 question regarding our claim that "unlike eQTL variants, sQTL variants tend to be located
 around intron splice sites rather than the canonical TSS of genes". One could hypothesise
 that these sQTL are enriched near alternative transcription start sites, but investigating this
 would require long read data and is outside the scope of our study.

Changes to manuscript:

We have now added the following text as well as the figure in this response as
Supplementary Figure 20. We have also removed any mention of PTPN2-205, except when
it's mentioned in Figure 6d as "the only annotated transcript that uses the implicated splice
junction":

*"Although the implicated splice junction exists in only one annotated transcript (PTPN2-205),*
*we found that it is used in at least four transcripts in a PBMC sample that was sequenced*
*using Pacbio's long-read sequencing (Supplementary Figure 20)."*

**5) The major claim of this study (as indicated by the title) is that**
**low-usage splice junctions underlie the risk of immune-mediated**
**diseases. However, only the percentage of IUR < 0.1 sQTL in the**
**entire dataset (53.4%) was presented. Could this proportion be**
**affected by stimulation?**

*Figure: proportion of colocalised splice junctions in each condition colored by intron usage ratio.*

We did not find evidence that the reported percentage is affected by stimulation. Across
conditions, we found that between 46.3% to 58.8% of colocalised splice junctions had mean
intron usage ratio (IUR) < 0.1, which is consistent with the overall reported percentage. More
specifically, the mean proportion of colocalised splice junctions with a mean intron usage

ratio less than 0.1 was 55.1% 52.4% for resting and stimulated cells, respectively. We have
added a sentence to the manuscript highlighting this fact.

**Changes to manuscript:**

We have added the following text to the subsection “Lowly-used alternative splicing events
underlie complex disease risk” and the figure in this response as Supplementary Figure 11:

*“We observed that 52.9% of colocalised sQTL introns have a mean intron usage ratio (IUR)
< 0.1 across samples (Figure 5a; 55.1% in non-stimulated conditions and 52.5% in
stimulated conditions; Supplementary Figure 11).”*

***6)The findings related to PTPN2-205 are crucial for supporting the***
***author's claim regarding the significance of low-usage splice***
***junctions. However, I have several concerns regarding the***
***interpretation of these results. Firstly, the sQTL SNP rs80262450***
***exhibits a strong eQTL signal for whole blood cells in the GTEx***
***data (it also has a strong eQTL effect for RP11-973H7.1 in colon***
***tissues). Therefore, the absence of colocalization observed for***
***eQTL in the macrophages under 13 different conditions does not***
***necessarily rule out the possibility that eQTL is the true causal***
***factor at this locus. Furthermore, the authors speculated that***
***increased usage of PTPN2-205 might attenuate the role of PTPN2***
***as a negative regulator (line 333). However, given that this***
***transcript consists of only three exons, it raises a question about***
***how it could function as a regulator. To address this issue,***
***experimental validation may be necessary.***

We don't consider the findings related to PTPN2-205 as crucial for supporting our claim
about the importance of low-usage splice junctions. Rather, our observation that over half of
our sQTLs relate to low-usage splice junctions is the main supporting evidence for our claim.
Such genome-/transcriptome-wide observations provide a much stronger evidence basis for
biological importance than could ever be drawn from a single locus. Among the colocalising
sQTLs for lowly used introns, we chose to highlight *PTPN2* simply because of its biological
relevance to IBD, rather than it being central to any genome-wide claim about low-usage
splice junctions.

We're aware that our colocalisation analysis showed evidence of colocalisation with *RP11-*
*973H7.1* eQTLs in several tissues including Colon_Transverse. In Colon_Transverse, the
IBD locus colocalised with both an *RP11-973H7.1* eQTL and a *PTPN2* sQTL. *RP11-973H7.1*
(renamed to *AP005482.1* in GENCODE v27) was not tested in MacroMap because it had
low expression levels (median TPM per condition=0-0.55) which were comparable to GTEx's

Colon_Transverse expression levels (median TPM=0.24). We cannot exclude the possibility
 that a *RP11-973H7.1* eQTL in MacroMap may have also colocalised with the IBD locus with
 high-confidence had it been more highly-expressed (a limitation of all molecular QTL
 mapping studies). Therefore, the association signals of both the *RP11-973H7.1* eQTL and
 *PTPN2* sQTL in Colon_Transverse suggest that either of these genes could plausibly
 underpin IBD risk at this locus. Given the long-range LD in the locus, it is difficult to
 statistically distinguish which QTL underpins the IBD risk without functional validation. We
 have better highlighted this fact in the discussion section of our manuscript.

Figure: Regional association plots for IBD (top), and *PTPN2* sQTL (middle; $PP_4=0.995$) *RP11-973H7.1* eQTL (bottom; $PP_4=0.97$) in Colon_Transverse. Colors indicate LD with the lead IBD SNP rs80262450.

Figure: Heatmap of colocalisation PP4 values in all GTEx tissue in all genes in the IBD-associated locus in 18p11.21. Colocalisation values with GTEx eQTLs are shown in the left heatmap and with GTEx sQTLs in the right heatmap. Colocalisation evidence exists for both a PTPN2 sQTL and a RP11-973H7.1 eQTL in multiple tissues including Colon_Transverse.

Multiple lines of evidence, including coding variants associated with monogenic IBD and mouse knock-out models have suggested *PTPN2* is the likely effector gene at 18p11.21. *PTPN2* has a known role as an inhibitor of the JAK-STAT pathway (JAK inhibitors are used to treat IBD), and mouse knockout studies have shown that the capacity of macrophages to clear bacterial invasion is impaired in *PTPN2*-deficient mice. Importantly, it is not yet known if and how common IBD-associated SNPs disrupt the function of *PTPN2*. We believe that our work throws new light onto the potential mechanism via which common IBD-associated variants perturb the function of *PTPN2* to increase risk of IBD, namely by dysregulated splicing of *PTPN2*. While it may seem implausible that a three-exon transcript may be translated to protein, Pacbio’s long-read RNA-seq data show that the implicated splice junction is present in multiple unannotated transcripts. We therefore refrain from making any conclusions about which specific transcript may be driving the disease association brought about by this sQTL. Experimentally following up our finding could be the means via which *PTPN2* is conclusively established as the effector gene at this locus. We believe this experimental work is beyond the scope of current paper, in part because it requires expertise we don’t have within our computational genomics group. Furthermore, our work has found hundreds of loci across a broad range of immune-mediated disease where splicing variation may impact disease risk. Each of these findings warrants functional follow-up in an appropriate and bespoke manner and thus we wish to publish our findings to expedite this.

Changes to manuscript

We have added the following text to our Discussion section and Supplementary Figure 23 to show the IBD-associated signal alongside the *PTPN2* sQTL and the *RP11-973H7.1* eQTL signals in GTEx’s Colon_Transverse

“...similar to other QTL studies, long-range LD in disease-associated loci makes the identification of effector genes challenging. This is exemplified by our *PTPN2* sQTL example,

*where an RP11-973H7.1 eQTL in several tissues also colocalises with the IBD-associated*
*signal. This makes it difficult to statistically distinguish which QTL underpins the IBD risk*
*without functional validation (Supplementary Figure 23)."*

**7)The GENCODE annotation used in this study might be too old**
**(v27) to precisely evaluate the "low-dosage" junctions. Would the**
**findings substantially change if the most recent annotation (v44)**
**were used? In particular, the author's claim that "Moreover, 50.6%**
**of these introns are not found in any annotated transcripts in**
**GENCODE v27, whereas only 12% of introns with a mean IUR ≥ 0.1**
**are absent from GENCODE v27 (Figure 6b)" should be based on the**
**recent annotation.**

We would like to thank the reviewer for this comment. Based on another reviewer's
comment, we have added Alzheimer's disease to the GWASs that were used for
colocalisation analysis. This led to a slight change in the percentage of colocalised splice
junctions that are unannotated, which was 49.87% for low-usage splice junctions versus
11.26% for common-usage splice junctions (IUR < 0.1 and IUR ≥ 0.1 , respectively). When
we used the latest GENCODE version (v45), these percentages changed slightly to 47.7%
and 11.01%.

We also acknowledge that the definition of 'unannotated' needs clarification. We consider a
splice junction to be unannotated if the exact combination of particular/acceptor splice sites
is not detected in GENCODE annotations. We have therefore investigated more thoroughly
the percentage of splice junctions that had a known donor/acceptor combination (DA), a
novel acceptor (D) or a novel donor (A) splice site, a novel combination of known acceptor
donors (NDA), or a novel acceptor and donor (N). In our Figure 5b, only DA is considered
"annotated". For the sake of completeness, we provide the number of colocalised splice
junctions in each category. We note that in only 0.07% of low-usage colocalised splice
junctions, both acceptor and donor splice sites were novel. 35.3% of splice junctions had
either a novel acceptor or a novel donor splice site, and 11.4% had a novel combination of
known acceptor/donor splice sites.

**Changes to manuscript**

We have now amended Figure 5 to be based on Gencode v45 and have changed the
percentage of unannotated introns to reflect those based on Genocode v45. A figure
showing all the splice junction classifications was also added as Figure 5d. Additionally, in
the sub-section "Lowly-used alternative splicing events underlie complex disease risk" we
have amended the text to clarify what we mean by "unannotated":

*"Despite the strong enrichment of unannotated introns in low-usage introns, only 0.7% of*
*low-usage introns had both a novel acceptor and a novel donor donor splice site. 46.7% of*

*low-usage introns feature either a known acceptor site only, a known donor site only or an*
*unannotated combination of known acceptor and donor sites (Figure 5d)."*

Minor points.

**8) Please show the location and structure of genes in Figure 7b and c.**

This was corrected and Figure 7b and 7c now show the location/structure of genes.

**9) I could not find rs80626450 (line 316) in dbSNP.**

This is a typo that was corrected to rs80262450.

Reviewer #2 (Remarks to the Author):

This manuscript examines splicing quantitative trait loci in macrophages exposed to various
cellular conditions. They report a large number of genes harboring sQTL effects, many of
which showed colocalization with immune mediated disease-associated risk loci. They
observed that about half of the colocalizations involved lowly-used splice junctions. An
example event in PTPN2 was shown. The findings support the potential role of alternative
splicing in immune mediated disease (IMD) risks. However, there are some major concerns
that need to be addressed.

***1) In comparing sQTL prevalence across conditions (Fig. 3c), power***
***is a major issue. The authors should do the comparison in the***
***context of a power evaluation. The events are comparable only if***
***they have similar power in different conditions.***

We would like to thank the reviewer for their comment. In Figure 3c, we compared the
presence of response sQTLs in RNA-seq samples following stimulation either at 6 hours, 24
371 hours, or both. We absolutely agree with the reviewer that the relative power to detect a
372 genetic effect needs to be considered when comparing across conditions. As such, we went
to great lengths to appropriately consider heterogeneity in effect size estimates across
conditions. Our definition of a response sQTL is based on mash (multivariate adaptive
shrinkage, Urbut et al, Nat Genet 2018), which tests if a QTL effect has a significantly
different effect size between a stimulated condition and a baseline condition (we define the
baseline condition as unstimulated macrophages; Ctrl), while accounting for uncertainty in
effect size estimates due to variable power. Mash computes a posterior effect size estimate
using a prior effect size distribution constructed from different data-driven and canonical
covariance matrices.

We declare an effect to be a response sQTL only if it shows significant evidence of deviating
from the prior distribution (i.e. if the posterior effect size estimate has a local false sign rate

LFSR < 0.05). This is intended to identify response sQTLs even in the presence of power
 differences between different conditions. In line with this, we found no correlation between
 the number of response sQTLs and RNA-seq sample size across different stimulation
 conditions ($R^2=0.02$). Thus, we are confident that our use Mash is the justified and enables
 388 us to make robust effect size comparisons between conditions even in the presence of
 389 differences in power.

 *Figure: the number of genes with response sQTLs is not correlated with RNA-seq sample size across conditions.*
 *Different stimulation conditions are shown in different colors and different timepoints are shown in different*
 *shapes. Blue line and shaded area represent the line of best fit and 95% confidence interval, respectively.*

**2) A major concern is with the main finding claimed by the study**
 **that many lowly observed splicing events may underlie IMD risks. It**
 **is well known that RNA-seq analysis may yield spurious spliced**
 **junctions, which may lead to identification of low usage splicing**
 **junctions that are indeed false positives. The authors need to carry**
 **out rigorous quality checks in read mapping and splicing event**
 **identification to rule out this possibility. Since this is a major**
 **conclusion of the paper (in the title), this issue needs to be**
 **investigated very thoroughly.**

 We very much agree with the review that we need to ensure our conclusions around the
 relative importance of low usage splice junctions are robust and not underpinned by quality
 control issues.

 We performed the following splice junction calling and QC steps to filter out potentially
 spurious splice junction calls:

1. We used WASP to filter out split reads whose mapping quality is affected by genetic variants. WASP first identifies reads that harbour genetic variants, then proceeds to re-map those reads to the genome with both the reference and alternative allele(s), and finally removes any reads whose mapping quality changes upon flipping genetic variant sites.
2. We used the following parameters in the splice event identification step:
 - A minimum and maximum intron length of 50 and 500,000 basepairs, respectively, to minimise the risk of too-short or too-long splice junctions.
 - A minimum of 50 split reads to identify an intron cluster
 - A minimum anchor length of eight nucleotides. Anchor length is different from overhang length in that it requires both ends of a splice junction to be covered by at least a specified number of basepairs.
3. Before mapping splicing QTLs we also removed introns with zero counts in > 40% of samples and introns with standard deviation < 5×10^{-3}

In addition, we gave further scrutiny to those sQTLs that colocalised with GWAS loci because we believe this set are the most likely to be functionally followed up by other researchers. We inspected the colocalised splice junctions with respect to three QC parameters: ratio of multi-mapping, alignment score, and splice site motifs. We expect these metrics to be similar for low-usage and common-usage splice junctions, and any deviation from this could indicate that our low-usage splice junctions are enriched with false-positive calls relative to the common-usage splice junctions. However, we found that low-usage splice junctions actually have slightly higher mean alignment scores than common-usage splice junctions (147.6 and 146.8 respectively; T-test P-value= 2.2×10^{-18} ; Figure), and that both sets have similar median proportions of multi-mapping split reads across samples (7.6% and 7.8% of intron-condition pairs have a median multi-mapping ratio > 0.5, respectively; Figure). Moreover, we found that an overwhelming majority of introns in both sets feature a canonical splice site GT/AG (99.1% and 99.2%, respectively). In order to validate the replicability of low-usage colocalised introns in other RNA-seq samples, we leveraged Intropolis, a database of introns identified using over 21,000 RNA-seq samples from the Sequence Read Archive. We found that only 1.5% of low-usage colocalised introns (N=12) were not present in Intropolis. Additionally, 92% of the low-usage colocalised introns were identified in at least 100 RNA-seq samples (Figure). Taken together, this evidence indicates that our low-usage colocalised introns are highly replicable and that there are no differences in various QC metrics between colocalised introns with low- and common-usage. We thus think our finding that many lowly observed splicing events may underlie IMD risk is robust.

Figure: Distribution of mean alignment score in low-usage and common-usage splice junctions. Mean alignment score was calculated per sample and then the mean was calculated across all RNA-seq samples per condition for each colocalised splice-junction/condition pair.

Figure: Distribution of median multi-mapping ratio in low-usage and common-usage splice junctions. Multi-mapping ratio was calculated per sample and then the median was calculated across all RNA-seq samples per condition for each colocalised splice-junction/condition pair.

Figure: Number of common-usage (top) and low-usage (bottom) splice junctions that feature canonical and non-canonical splice site motifs. Splice site motif information were obtained the *.SJ.out.tab file that are output by the STAR aligner.

Figure: Distribution of the number of Intropolis Sequencing Read Archive RNA-seq samples in which colocalised low-usage introns can be detected.

Changes to manuscript

The following texts has been added to sub-section “Lowly-used alternative splicing events
underlie complex disease risk”:

• “Furthermore, we did not observe differences in alignment score or multi-mapping
rates of split reads between common-usage and low-usage introns”

• “In order to validate the replicability of low-usage colocalised introns in other RNA-
seq samples, we leveraged Intropolis, a database of introns identified using over
21,000 RNA-seq samples from the Sequence Read Archive. We found that only
1.5% of low-usage colocalised introns (N=12) were not present in Intropolis.
Additionally, 92% of the low-usage colocalised introns were identified in at least 100
RNA-seq samples”

• The figures above has been added to the manuscript as Supplementary Figure 12-
14,17.

• Additionally, we have decided to change the title of our manuscript to “Splicing QTL
mapping in stimulated macrophages associates low-usage splice junctions with
immune-mediated disease risk”. The new title reflects our uncertainty around whether
low-usage splice junction underpin IMDs or whether they simply reflect decreased
splicing efficiency that results in increased erroneous splicing. Additionally, we have
discussed this possibility in our “Discussion” section.

**3) The PTPN2 example also brought up a question about effect size**
**of the sQTLs. It seems the different genotypes have very small**
**differences in their IUR. With such a small splicing change, it is not**
**reasonable to assume a functional impact exists. This undermines**
**the premise for the entire paper. The authors should examine effect**
**sizes of sQTLs, and apply a requirement on minimum effect sizes**
**that make biological sense.**

We would like to clarify that, like the majority of QTL mapping studies, we have performed a
rank-based inverse normal transformation step before we mapped sQTLs (which is achieved
by using the --normal option in QTLtools). This is a necessary step when mapping QTLs,
which use a linear regression model to detect associations between the genotype and a
given quantitative phenotype. This normalisation step has important implications for the
interpretation of effect sizes. Most importantly, the resultant effect size estimates measure
the change in *normalised* phenotype (i.e. normalised intron usage ratio) per one unit of
genotype change. This means that the effect size estimate is not dependent on the absolute
value of its usage ratio (e.g. an sQTL effect that leads to a mean IUR change from a
normalised mean IUR value of -1 in homozygous reference individuals to 1 in homozygous
alternative individuals will have the same effect size estimate regardless of absolute mean
IUR difference between genotype groups). Consistent with this, we found that the effect size

estimates were similar for both common-usage and low-usage colocalised splice junctions
(Figure). Therefore, our statistical power to detect sQTL effects is a function of 1) the minor
allele frequency 2) the effect of the SNP on the normalised usage ratio, rather than the
absolute value of the usage ratio.

*Figure: Absolute effect size for common-usage (orange) and low-usage (black) splice junction showing that the*
*effect size(x-axis) is not dependent on absolute value of IUR (y-axis).*

It is also important to note that common variant (MAF > 5%) effects discovered via GWAS
and QTL studies tend to be modest. Deleterious mutations with large effect sizes tend to be
rarer in the population due to purifying selection. Interestingly, in plants, the majority of
sQTLs that were found to colocalise with important traits showed small isoform usage
changes without involving major isoform switching between genotypes [4]. We thus
respectfully disagree with the premise that a small change in splicing cannot have an
important impact on delicately balanced and complex biological systems. Furthermore, the
biological consequences of splicing variation are incompletely understood, and thus it is not
possible to somehow restrict our transcriptome-wide analyses to those that 'make biological
sense' (even putting aside that fact that this would vary from gene to gene and our analyses
are based on a rank-based inverse normal transformation of the RNA-seq data).

**4) For sQTL introns with a mean intron usage ratio (IUR) <0.1, what**
**are the IUR values for each genotype? What are the effect sizes?**
**What are the power in detecting them as significant sQTL? These**
**data should be shown in detail. For each intron, are all genotypes**
**associated with low IUR, or certain genotypes with relatively high**
**IUR (e.g. >0.1) and others with low values? If it is the latter case,**
**then it does not make much sense to call them lowly used.**

A mean intron usage ratio < 0.1 indicates that across all individuals the mean intron usage is
less than 0.1. As this is a mean across all individuals in the study, we believe it is
appropriate to refer to them as low usage intron ratios. For an sQTL to be detected there
must be a significant difference in mean intron usage with respect to genotype. Figure 6b
shows an example of this for the PTPN2 sQTL. Here, one can see that while there is
variation within genotype with regard to normalised intron usage ratio, there is a significant
increase in mean normalised usage with increasing number of copies of the alternative allele
at rs80262450. Across our low usage sQTLs, there may well be some individuals where the
mean intron usage is greater than 0.1, especially if the mean intron usage ratio across all
individuals is just less than 0.1, there is a lot of variance in intron usage between individuals
and the genetic effect on this variance is large. Thankfully, the sQTL mapping approach we
have adopted takes these variables into account when assessing the significance of genetic
effects on intron usage. Given we find a total of 5,734 genes with significant sQTL effects
and 707 genes with high-confidence colocalisations, it is simply not possible to report the
intron usage values for each genotype (nor is it really relevant because, as mentioned
above, we perform our tests following a rank-based inverse normal transformation). The
power to detect sQTL effects is also variable depending on the sample size, minor allele
frequency of variants being tested, the variance in intron usage ratio across individuals and
the variance explained by genetic effects. Reassuringly, robust molecular QTLs have been
successfully and replicably identified in sample size much smaller than ours. Furthermore,
our robust approach to QC and controlling for multiple testing gives us confidence that the
majority of our reported sQTL effects are real.

**5) Experimental validations are needed to confirm the existence of**
**true lowly used splicing isoforms, as this is a major conclusion of**
**this paper.**

We agree with the reviewer that experiment validation is required to test the hundreds of
hypotheses that our paper has generated with regards to the relationship between individual
low usage splice junctions and risk of a common complex disease. Each of these
experiments will require specific expertise relating to the particular gene/pathway and
disease. This expertise is not available within our lab, nor is it possible for all of these
hypotheses to be tested by a single lab. Our aim is to rapidly publish our results so that labs
with the expertise in the particular genes and pathways can begin functionally following up
our hypotheses. This is in keeping with the standard approach for molecular QTL mapping
studies, which typically identify thousands of genetic effects.

We do agree with the reviewer that it is necessary for us to do all we can confirm the
existence of lowly used splice isoforms. In line with the expertise of our group, we have
undertaken various quality control steps to maximise the robustness of our results and
sought to validate our results using orthogonal datasets. This work is outlined in detail in our
response to the third comment from this reviewer. In summary, we found that low-usage
splice junctions actually have slightly higher mean alignment scores than common-usage
splice junctions, and a similar median proportion of multi-mapping split reads across
samples. We found that an overwhelming majority of introns in both sets feature a canonical
splice site GT/AG (99.1% and 99.2%, respectively). We found that only 1.5% of low-usage
colocalised introns (N=12) were not present in Intropolis, a database of introns identified
using over 21,000 RNA-seq samples from the Sequence Read Archive. Additionally, 92% of
the low-usage colocalised introns were identified in at least 100 RNA-seq samples (Figure).
Taken together, this evidence indicates that our low-usage colocalised introns are highly
replicable and that there are no differences in various QC metrics between colocalised
introns with low- and common-usage. While a small percentage of lowly usage splicing
events could well be false-positives, we believe our finding that many lowly observed splicing
events underlie IMD risk is robust to this.

**6) In the *PTPN2* example, it is not clear if the 205 isoform is actually**
**expressed because Fig. 6d does not seem to show read distribution**
**overall the first exon of this isoform which is unique to 205. This**
**needs to be clarified and experimental validation should be**
**conducted to confirm its expression.**

We thank the reviewer for their comment. As outlined in the previous comment, we are not
confident that the implicated *PTPN2* splice junction is necessarily unique to *PTPN2*-205. We
will refrain from making such a conclusion in the paper. Nonetheless, we have provided a
clearer RNA-seq coverage plot upstream of the donor splice site chr18:12,817,365 as
demonstrated by a small number of exonic reads (bars) and split reads (arcs).

Figure: RNA-seq coverage plot showing mean number of reads that map to PTPN2 exons (y-axis), the canonical PTPN2 transcript structure and the PTPN2-205 transcript structure. Bottom plot show RNA-seq coverage shows exonic reads mean coverage in exon 1 of PTPN2-205. Each coverage plot is stratified by individuals who have 0, 1, or 2 copies of the alternative allele of rs80262450.

**7) Page 14, line 376-378, the conflicting directions of effect in**
**different conditions may be interpreted as weak and**
**unreproducible results, rather than context-dependent results. It is**
**necessary to consider this and prove they are not false positives.**

We thank the reviewer for this suggestion, which has proved extremely useful. Upon further
exploration, we noticed that we had mistakenly included in this analysis lead sQTL effects for
a given gene-condition pair that were often independent of effects detected for other
conditions at that gene. Independent effects at SNPs in low linkage disequilibrium cannot be
used to make robust inferences about the sharing of a single genetic effect across
conditions. We have now removed the statement previous on Page 14, lines 376-378 and
removed Supplementary Figure 6. We thank the reviewer again for spotting this error.

**Minor:**

**Figure 2a, b: consider providing more detailed labels to show both conditions and**
**timepoints, it is currently very hard to read.**

We have now added a caption to Figure 2 to clarify what each acronym in the figure stands
for.

**The variants with colocalization signals are not necessarily causal. Please avoid**
**using such a strong term.**

We agree that variants with colocalisation signals are not necessarily causal. Rather,
colocalisation analyses quantify the extent to which two genetic effects in a region are likely
to be driven by the same (often unknown) causal variant [1]. Without undertaking statistical
fine-mapping and functional follow-up analysis it is not possible to state which of the variants
in high-linkage disequilibrium with the lead SNP(s) is causally underpinning the shared
association signal. We have taken care in our manuscript to not imply that the leads SNPs
from the sQTL, eQTL or disease GWAS study are the true causal variant. Rather, we use
the term causal variant in an abstract sense to describe a variant that underpins both the
molecular QTL effect (sQTL or eQTL) and the disease association. This approach is in
keeping with common practice within the field of statistical genetics and, therefore, for clarity
we would like to stick with it. If the editor deems it necessary we could instead use a term
such as “underlying variant” but we fear this would just cause confusion.

**The Results section contained extensive discussions. Suggest moving those to**
**Discussion.**

**Changes to manuscript**

We have now removed three paragraphs that we, in agreement with the reviewer, think are
more appropriate for the Discussion section. These were previously part of the Results
sections:

- 1. “Macrophage stimulation initiates an alternative splicing programme in innate
immune response genes” (“*Our differential splicing analysis reinforces the role of*
*alternative splicing in regulating important innate immunity processes...*”)

- 2. “Splicing QTLs identify additional GWAS loci effector genes not detected by
expression QTLs” (“Recently, Mountjoy et al. 2021 (ref: 32) colocalised 50.7% of
tested GWAS loci...”)
- 3. “Lowly-used alternative splicing events underlie complex disease risk” (“Our
observation that over half of colocalised sQTL introns are lowly used...”). We believe
the messages of these two paragraphs are already reiterated in the Discussion
section.

Reviewer #3 (Remarks to the Author):

This manuscript studies how genetic variants are associated with alternative splicing in
macrophages derived from iPSCs. They focus on how splicing QTLs may be contributing to
immunological diseases by using GWAS data. The authors developed a macrophage
activation map where they stimulated macrophages with ten different stimuli at two different
time points. They demonstrate how alternative splicing varies among specific stimuli,
particularly in innate immune response genes. They observed that sQTLs that co-localize
with GWAS loci often involve low abundance introns. They provide specific examples of their
findings. For instance, they discovered an sQTL for the PTPN2 gene that is specific for
activation conditions and that implicates a low abundance intron. This sQTL colocalizes with
an IBD risk locus. Additionally, they identified two other colocalization signals for LRRK2 and
DENND1B in IBD, which participate in the same pathway. In summary, the authors report a
novel macrophage sQTL map, aiding in the understanding of immune-mediated diseases
and highlighting the importance of splicing and cell state context.

The manuscript is well written and well structured. The data used in this manuscript were
previously generated and constitute a very large collection of useful data for the field. A
focus on the splicing aspect of this dataset is insightful and worthy of publication. However,
several of the analyses feel a bit “rushed” and I believe deeper and more thorough analyses
are necessary to support the main claims of the paper.

Major concerns

***1. Replication of macrophage sQTLs. In most QTL studies people
check if their QTLs replicate compared to other QTL studies in the
same cell-type. Given these are iPSCs derived macrophages, I
believe replication in this study would be of particular relevance. It
is important to know how similar the QTLs found in this study are
to primary macrophages. There are multiple macrophage QTL
studies in eQTL Catalogue from which summary statistics could be
easily used, given they also used LeafCutter for splicing
quantification***

We would like to thank the reviewer for this helpful suggestion. We downloaded two
macrophage eQTLcatalogue datasets that quantify alternative splicing using Leafcutter

(Alasoo et al. 2018 and Nedelec et al. 2016). To assess replication, we used the lead SNP
 701 per intron and measured their replication between MacroMap and each external dataset
 using two metrics: replication rate (π_1) and correlation of effect sizes. Importantly, we limited
 our replication to naive macrophages, both in MacroMap and in external datasets.
 Stimulated conditions in these studies either did not match our stimulant panel or our
 stimulation timepoints (Alasoo et al. 2018 profiled naive macrophages, Salmonella_5h, or
 IFNG_18h + Salmonella_5h; Nedelec et al. 2016 had naive, Listeria_5h and
 Salmonella_5h). We noted both high replication rates and high correlation between effect
 size estimates ($\pi_1 > 0.7$; pearson correlation coefficient $\rho > 0.9$).

 *Figure: Replication of lead SNP per intron between MacroMap Ctrl_6/Ctrl_24 and naive macrophage sQTLs from*
 *Alasoo et al. 2018 and Nedelec et al. 2016. Both datasets were preprocessed as part of eQTL catalogue.*

Changes to manuscript

The figure in this response was added as Supplementary Figure 4 and the following text was
 added to the subsection “Macrophage stimulation increases the number of genes with
 significant sQTL effects”:

 *“We noted both high replication rates and high correlation between effect size estimates for*
 *our sQTLs and those from two macrophage datasets from the eQTL Catalogue ($\pi_1 > 0.7$;*
 *pearson correlation coefficient of effect sizes $\rho > 0.9$; Supplementary Figure 4).”*

**2. Colocalization analyses.**

***a. The authors use different PP4 thresholds depending on their***
***analyses, with little information on the rationale behind. Especially***
***for considering an sQTL-only colocalization with GWAS, they***
***require sQTL PP4 > 0.75 and eQTL PP4 just needs to be lower than***
***0.75. I imagine there could be a lot of cases where eQTL PP4 is in***
***the range of 0.65-0.74, and in that case they are not considered to***
***colocalize with GWAS, when in an earlier analysis they used***
***PP4 > 0.1 as a threshold of colocalization. The current threshold***
***could be over-estimating how many cases are sQTL-only loci. Have***
***the authors explored how the results change at other thresholds or***
***in a more continuous manner?***

We thank the reviewer for their comment. First, we agree that the rationale behind different
PP4 thresholds is not clear in the main text. We would like to clarify what guided the different
cutoff choices.

a) We used a PP4 threshold of 0.75 to declare a shared causal variant driving a
742 molecular QTL (sQTL or eQTL) and a GWAS hit because we wanted to
743 somewhat conservatively define colocalisation events defining potential
disease effector genes. We used a threshold of $PP4 \geq 0.75$, which is
consistent with existing literature.

b) When choosing specific examples of genes with sQTL signals that colocalise
with an IMD locus to highlight in the manuscript, we used a more stringent
PP4 threshold of 0.9 (e.g. in the *PTPN2* example). We did so because we
wanted to ensure we focus on the strongest colocalisation examples.

c) We used a PP4 threshold of 0.1 to declare a shared causal variant driving an
751 eQTL and sQTL effect because we wanted to generously define
colocalisation events and provide an upper bound for the degree of sharing.
Even at an extremely permissive PP4 threshold ($= 0.1$), very few significant
sQTL signals colocalise with an eQTL signal for the same gene in the same
condition.

We agree with the reviewer that there will be loci where the PP4 between the eQTL and a
GWAS signal is in the range 0.65-0.74 and therefore a colocalisation would not be declared.
This is a common problem in statistics when using thresholds to define 'winners'. To explore
how much our choice of PP4 threshold for declaring colocalisation events affects our
conclusion, we undertook the sensitivity analysis proposed by the reviewer. We varied the
eQTL PP4 threshold while maintaining the sQTL PP4 threshold at 0.75 (i.e. $sQTL\ PP4 \geq$
0.75 and $eQTL\ PP4 < k$, where $k=0, 0.05, \dots, 1$). The expectation is that with a permissive
(low) threshold k , a very small percentage of loci will colocalise solely with an sQTL, but as k
increases so will the percentage of colocalisation driven solely by an sQTL. We also
undertook the inverse analysis, varying the sQTL PP4 threshold while maintaining the eQTL
PP4 threshold at 0.75 (i.e. $eQTL\ PP4 \geq 0.75$ and $sQTL\ PP4 < k$, where $k=0, 0.1, \dots, 1$). The

results of this analysis are shown in the figure below. The results of this analysis were in line
with our expectation described above but, for a given PP4 threshold, the proportion of loci
mapping solely to an sQTL was greater than the proportion of loci mapping solely to an
eQTL.

*Figure: Percentage of loci that colocalise solely with an sQTL at PP4 threshold ≥ 0.75 and using different PP4*
*threshold for eQTL colocalisation (top). Percentage of loci that colocalise solely with an eQTL at PP4 threshold*
*≥ 0.75 and using different PP4 threshold for sQTL colocalisation (bottom).*

Changes to manuscript

- ● The figure in this response was added as Supplementary Figure 10 and the following
text was added to the subsection “Splicing QTLs identify additional GWAS loci
effector genes not detected by expression QTLs”

*“The contribution of loci that colocalised solely with an sQTL remained larger than the*
*contribution of loci that colocalised solely with an eQTL across a range of exclusion*
*PP4 cutoffs (Supplementary Figure 10)”*

- • We now justify the usage of each PP4 cutoff:
1. *“only 25% of sGenes likely share a single causal variant with an eGene in the*
*same condition even with a permissive colocalisation cutoff (PP4 ≥ 0.1; ...”* in
the significant sQTL and eQTL colocalisation.
2. *“(with a strict PP4 cutoff ≥ 0.9)...”* in the specific colocalisation examples

***b. Given the authors are claiming that sQTLs play such an***
***important role in complex traits, have the authors considered***
***performing colocalization analysis with more stringent methods,***
***such as JLIM (Chun et al. Nature Genetics, 2017)? In the 21 IMDs***
***selected, the authors found that close to 75% of risk loci are***
***explained by co-localization with e/sQTLs of their macromap, while***
***several of these top IMDs are known to likely have a different main***
***cell-type mediating disease risk. For example, in studies assessing***
***enrichment of heritability in cell-type specific annotations, PBC***
***appears more of a B cell driven disease, MS is considered a T cell***
***and B cell driven disease, and for childhood-onset asthma they***
***have reported a T cell component. Can the authors comment on***
***this results (which in my opinion are unexpected)? Perhaps the***
***authors can include diseases that are known to have a strong***
***macrophage/myeloid component (such as Alzheimer disease,***
***Gjoneska et al., Nature, 2015), and some “negative control”***
***diseases that are expected to not have an immune component,***
***such as psychiatric diseases.***

We thank the reviewer for their comment. It is difficult to comment on particular disease
colocalisations and how they relate to other non-myeloid cell types within the scope of our
analyses. Several factors could account for the unexpected high colocalisation yields for
some diseases in our study. First, sQTL effects could be shared between several immune
cell types, including T-cells and B-cells. Second, it could be the case that heritability for a
given disease is enriched in *several* cell types. For these reasons, we adopted an approach
that compares groups of diseases, rather than individual diseases. We compared the
colocalisation yield of immune-mediated diseases versus psychiatric and cognitive traits or
disorders. We repeated our colocalisation analysis with 11 cognitive traits, and we found
that, on average, 34.7% of loci colocalised with Macromap sQTLs, eQTLs or both, a
considerably lower rate than that of immune-mediated disorders (52.6%).

Figure: Proportion of neuro-psychiatric disorder loci that colocalise with an eQTL, an sQTL or both (PP4 ≥ 0.75). Dotted blue line indicates the mean proportion of loci that colocalise with a Macromap eQTL, sQTL or both.

Changes to manuscript

The figure above was added as Supplementary Figure 9 and the following text was added to
the subsection “Splicing QTLs identify additional GWAS loci effector genes not detected by
expression QTLs”:

*“We then compared how many tested loci colocalised with each type of molecular QTL and*
*found that 52.6% (820/1,558) of tested loci were likely to share a single underlying variant*
*with either an eQTL, sQTL or both. This colocalisation yield was higher in IMDs than in a set*
*of 11 psychiatric and cognitive traits (34.7%; Supplementary Figure 9).”*

***c. In the methods the authors specify that in their colocalization***
***analyses they are including any locus that shares at least 2 SNPs***
***between the GWAS and QTL study. This threshold seems quite lax,***
***would the results change if the authors require a higher number of***
***shared SNPs in the locus in order to test for co-localization with***
***more information? (e.g. 20).***

We previously set the minimum requirement for the number of SNPs solely for the technical
reason that colocalisation cannot be conducted with a single SNP in common between two
traits. We agree that it is crucial to ensure that the conclusion about the higher colocalisation
yield of sQTLs is not affected by colocalisation events with too little overlap in SNP coverage
between the GWAS and the sQTL variants. We found that previously only 3% of colocalised
gene/trait pairs (46 gene/trait pairs) share 20 SNPs or less. Nonetheless, we have now
decided to remove those colocalisation events.

Changes to manuscript:

All relevant figures and numbers in the Results subsections “*Splicing QTLs identify*
*additional GWAS loci effector genes not detected by expression QTLs*” and “*Lowly-used*
*alternative splicing events underlie complex disease risk*” were regenerated. We have
amended the text in the Methods subsection “colocalisation analysis”:
“*We perform colocalisation analysis using the R package coloc v5.1.1 and we discard GWAS-*
*intron pairs that have < 20 variants in common.*”

**3. Depth of analyses for stimulation conditions.**

**a. It would seem only a couple of the stimuli were analysed in a**
**focused way. For the pathway enrichment analysis, only the**
**condition sLPS_6 and the PIC_6 were described in the results**
**section. How do the results of pathway enrichment look the other**
**stimuli? Do you get the same pathways enriched across stimuli or**
**do they change, and are the differences expected given the**
**activation conditions?**

We would like to thank the reviewers for their comment. We performed pathway enrichment
analysis using the REACTOME 2016 database, where biological pathways are organised in
a hierarchical structure. Each broad high-level pathway includes lower-level pathways with
increasingly specific pathway categories. Although we found differences in pathway
enrichment between different conditions and timepoints at low-level pathways (e.g. RIG-
I/MDA5 pathway in macrophages stimulated with viral mimic PolyI:C but not in macrophages

stimulated with bacterial stimulant sLPS), we noted that a relatively small number of
 conditions faithfully capture all the high-level enriched pathways (level-2 pathways).

For conditions where RNA is harvested six hours following stimulation, macrophages
 stimulated with sLPS, either alone or with other stimulants (i.e. LPS+IL10 or LIL10 and
 CD40+IFNG+LPS or CIL) capture all the level-2 pathways enriched across all 6-hour
 conditions (Supplementary Note Figure 1 and Supplementary Figure 2). Almost none of the
 other 6-hour conditions have any enriched level-2 pathways, suggesting that stimulation with
 LPS captures most of the pathways that are subject to differential splicing six hours following
 macrophage stimulation. This is consistent with a rapid response that macrophages exhibit
 towards activation with bacterial stimulants. On the other hand, viral stimulants and anti-viral
 cytokines tend to elicit a late response. Stimulation with PolyI:C and IFN β captures the
 same set of differentially spliced pathways after 24 hours (Supplementary Note Figure 2). It
 is expected that PolyI:C and IFN β have similar effects as PIC is known to lead to a strong
 but delayed production of IFN β and Interferon-stimulated genes (ISGs) [5].

 *Supplementary Note Figure 1: Level-2 REACTOME pathways enriched in differentially spliced genes across*
 *different stimulation conditions. Coloured labels on the left-hand side indicate the level-1 pathways that each*
 *pathway belongs to. After six hours, stimulation with CIL, LIL10 and sLPS captures all the level-2 pathways*
 *enriched across any of the six-hour conditions. After 24 hours, stimulation with PIC and IFN β captures all the*
 *level-2 pathways enriched across any of the 24-hour conditions. Double asterisks indicate significant enrichment*
 *(P -value $< 2.5 \times 10^{-3}$ for 20 conditions) and single asterisks indicate nominal enrichment (P -value < 0.05).*

It has been previously shown that while the macrophage response to LPS and PIC/IFN β
 implicates overlapping pathways, response to PolyI:C and IFN β is not immediate, while
 response to LPS starts within a few hours and subsides after 12 hours. We therefore
 conclude that stimulation with LPS and PolyI:C or IFN β capture most differentially spliced
 pathways but at different timescales.

 In this Supplementary Note, we now highlight some of the key high-level pathways that are
 enriched in differentially spliced genes:

**Cytokine Signalling and Trafficking Pathways**

Cytokine signalling pathways included pro-inflammatory cytokine pathways such as Toll-like
receptor signalling pathways, interferon signalling pathways, signalling by interleukins. LPS
binds to TLR-4 which initiates the MyD88-dependent signalling cascade that leads to the
production of pro-inflammatory cytokines. Additionally, a TRIF-dependent signalling pathway
leads to the production of Type I interferons (reviewed in [6]). Similarly, PIC is a potent
activator of TLR-3 leading to a TRIF-dependent activation of Type I Interferons. Our enriched
pathways recapitulate the interferon and interleukin signalling pathways (Figure) [7-9].
Additionally, membrane trafficking pathways were enriched for differentially spliced genes in
the same conditions. Membrane trafficking is crucial for a variety of functions including
trafficking pathogens and pro-inflammatory cytokines produced in response to stimulation. In
macrophages, membrane trafficking is a central component of cytokine production. As
macrophages lack secretory granules, which in other cell types are used to store cytokines
for rapid release [10], macrophages rely on membrane trafficking pathways to export
cytokines as they are released [11,12].

**Apoptosis**

Pathways related to apoptosis and cellular death were also enriched in differentially spliced
genes. Apoptosis plays central roles in pathological processes such as atherosclerotic
plaque formation. In the early stage of plaque formation, classically-activated M1
macrophages produce inflammatory cytokines that lead to increased inflammation in
response to the accumulation of lipoproteins. In later stages, macrophage apoptosis plays
an important role to limit the accumulation of inflammatory cytokines produced by
macrophages (reviewed in [13]).

**Splicing auto-regulation**

Interestingly, within the broad pathway "Processing of Capped Intron-Containing Pre-
mRNA", the "mRNA splicing pathway" was significantly enriched in sLPS_6, CIL_6, PIC_24,
IFNB_24. Differentially spliced genes include genes essential for the splicing of nascent
mRNA such as heterogenous nuclear ribonucleoproteins (hnRNPs L, M, K, UL1, F, A2B1, D,
and C), and splicing factors (SRSF11, SRSF5 and SRSF3 and SF3B1). It has been previously
shown that the overall levels of splicing regulators are often auto-regulated by adjusting their
own splicing to increase the production of unproductive isoforms (splicing auto-regulation;
reviewed in [14]). These results suggest that this mechanism may also be engaged upon
macrophage exposure to pathogens such as LPS and PIC and inflammatory cytokines such
as IFNB.

Supplementary Note Figure 2: Level-3 REACTOME pathways enriched in differentially spliced genes across different stimulation conditions. Coloured labels on the left-hand side indicate the level-1 pathways that each pathway belongs to. Double asterisks indicate significant enrichment (P -value $< 2.5 \times 10^{-3}$ for 20 conditions) and single asterisks indicate nominal enrichment (P -value < 0.05).

Changes to the manuscript

We added the level-2 enrichment figure as Supplementary Figure 2 and added a
 Supplementary note with the previous text. Additionally, we added the following text to the
 sub-section “Macrophage stimulation initiates an alternative splicing programme in innate
 immune response genes”:

*“At a high-level, enriched pathways re-capitulated known macrophage responses to*
 *pathogens, including enrichments of cytokine signalling, membrane trafficking, TCA cycle*
 *and metabolism of amino acids, responses to stress and apoptosis pathways*
 *(Supplementary Figure 2; Supplementary Note).”*

***b. The authors focus their analyses on comparing non-stim***
 ***condition with 6hr or 24 hour activation, but little is shown***
 ***regarding differences between activation conditions. Can the***
 ***authors comment on activation condition-dependent sQTLs? How***
 ***important is it to stimulate different pathways to capture disease***
 ***relevant sQTLs or would one activation condition capture most of***
 ***the disease associated effects?***

 We thank the reviewer for their comment. Following the reviewers’ suggestion, we have now
 evaluated how many conditions a particular high-confidence colocalisation event (defined as
 intron-trait pair) was detected in. We found that across diseases, an average of 68% of
 colocalisation events were colocalised with high confidence (PP4 ≥ 0.75) in only one
 condition, while the rest were colocalised in multiple conditions (Figure).

*Figure: Heatmap showing the number of high-confidence colocalisation events ($PP4 \geq 0.75$) where there is*
*colocalisation evidence in one condition and in more than one condition.*

However, this large number of single-condition colocalisations should not always be
interpreted as condition-specific events. To identify condition-specificity with high-
confidence, we used mash. mash shows that 8%-41.4% of colocalisation events per disease
were considered response sQTLs (Figure).

*Figure: Proportion of colocalisation events that are response sQTLs in different diseases.*

The previous two figures show two seemingly contradictory findings. On the one hand, it
seems that most colocalisations per disease can be detected in only one condition. On the
other hand, most colocalisation events per disease are not considered response sQTLs.
This discrepancy can be explained by how response sQTLs are defined by mash on one
hand and how colocalisation is assessed on the other. mash considers an sQTL effect to be
a response sQTL if its effect size is significantly different from the baseline effect size (in our
case resting macrophages Ctrl_6 and Ctrl_24). Colocalisation assesses the likelihood of two
association signals sharing the same single underlying variant. Therefore, differences in
effect sizes between conditions may result in inconsistent colocalisation evidence between
conditions, but these differences may not be large enough for the sQTL effect to be
considered a response sQTL effect by mash. From this, we conclude that practically, our
high colocalisation yield would not have been possible without the large number of
stimulation conditions we profiled. But biologically, our mash analysis shows that a smaller
proportion is likely to be response-specific.

Changes to manuscript:

We have now added the two figures in this response as Supplementary Figure 7 and 8 and
added the following text to the Results subsection “Splicing QTLs identify additional GWAS
loci effector genes not detected by expression QTLs” summarising the conclusion made in
this response:

*“On average, 68% of sQTL-disease colocalisation events could only be detected in a single*
*condition (Supplementary Figure 7) . However, a smaller percentage (25%) showed*
*evidence of being response sQTLs (LFSR < 0.05; Supplementary Figure 8). This suggests*
*that profiling a large number of stimulation condition results in a larger disease colocalisation*
*yield, but a smaller percentage of sQTL effects is likely to be response sQTLs.”*

***C. I would also like to see in the text more details about the stimuli***
***applied to the macrophages, and the rationale behind. A bit of that***
***is shown in the scheme of Figure 1, but nothing is mentioned in the***
main text, and it is an important component of the study.

Changes to manuscript

The following text was added to subsection “Macrophage stimulation initiates an
alternative splicing programme in innate immune response genes” of the Results:
*“Our stimulants were selected to cover a wide range of innate immune exposures.*
*This includes pro- and anti-inflammatory cytokines (IFN β , IFN γ , interleukin-4/IL4),*
*synthetic viral mimics (Resiquimod/R848, Poly I:C/PIC), bacterial mimic LPS with*
*and without other inflammatory cytokines (Pam3CSK4/P3C, CD40 ligand + IFN γ +*
*sLPS/CIL, interleukin-10 + sLPS/LIL10), and myelin basic protein (MBP) to mimic*
*brain-resident macrophage response to stimulation. In total, this resulted in 4,698*
*RNA-seq libraries across all 24 conditions.”*

**4. Lowly-used alternative splicing events and disease risk.**

***a. The authors show in Fig. 5 the “distribution of mean intron usage***
***ratio for colocalised introns, showing a peak close to 0”. How does***
***the distribution of intron usage ratio look for introns in general? Is***
***there really a bias towards low usage in colocalized sQTLs or is***
***that how the expected distribution look like?***

To test the enrichment of colocalised introns in low-usage introns, we compared the intron
usage distribution of colocalised intron-condition pairs (N=4,401) to all intron-condition pairs
that passed QC (N=2,032,481). We tested this enrichment at several mean IUR thresholds

(0-0.1) using a Fisher's exact test, and did not find consistent evidence of enrichment or
depletion across the range of mean IURs that we tested (Fisher's exact odds ratio=0.85-1.18
across mean IUR cutoffs; Figure and Figure). This means that the proportion of low-usage
splice junctions is similar in both quantified and colocalised splice junctions.

*Figure: cumulative distribution of mean intron usage ratio for all post-QC introns (grey) and colocalised introns*
*(red)*

 *Figure: Enrichment test for low-usage colocalised introns based on different mean IUR cutoffs for defining low-*
 *usage introns. Fisher's exact test odds ratio for the enrichment of colocalised low-usage introns is shown on the*
 *y-axis and mean intron usage ratio used to define low-usage introns is shown on the x-axis. An odds ratios < 1*
 *indicates depletion of low-usage introns among colocalised introns and an odds ratio > 1 indicates enrichment of*
 *low-usage introns among colocalised introns.*

**Changes to manuscript:**

We added the two figures as Supplementary Figure 15 and 16 and added the following text
 to the sub-section "Lowly-used alternative splicing events underlie complex disease risk":
 "We then asked if the set of colocalised introns is enriched for low-usage introns compared
 to all tested introns. We found no consistent evidence of enrichment across a range of IUR
 cutoffs (Fisher's exact test odds ratio=0.85-1.18 across mean IUR cutoffs=0.01-0.1;
 Supplementary Figure 15 and 16). This suggests that the large proportion of colocalised
 introns with mean IUR < 0.1 is a result of testing a set of introns with a similarly large
 proportion of low-usage IURs."

***b. The authors chose to consider only one intron usage per sQTL***
 ***gene, choosing the one with the lowest P-value. Can choosing only***
 ***the intron with lowest P-value also bias the results? Have the***
 ***authors tried to include sQTLs of other introns of the same gene?***
 ***Would those "other" introns that colocalize with GWAS also introns***
 ***of low usage? Or are introns with lowest P-value per gene biased***
 ***towards those of lowest usage?***

 We thank the reviewer for their comment. We would like to clarify a distinction between two
 major analysis we performed. First, we performed a permutation-based QTL mapping
 analysis where we discovered genes with significant sQTL effects. In order to correct for
 thousands of genes being tested, we perform an FDR correction, where one intron per gene
 is reported. This is not the same approach we use for colocalisation analysis. In our
 colocalisation analysis, we performed statistical colocalisation between GWAS loci and all

quantified introns that passed the sQTL QC. Both common-usage and low-usage introns are
thus tested for colocalisation.

***c. Are these lowly abundant alternative splicing events implicated***
***in disease enriched in non-sense mediated decay, as it is shown in***
***the paper: Fair et al, bioRxiv, 2023? How many of these lowly-used***
***alternative splicing events are likely to be translated into protein?***
***Can you validate experimentally some of them?***

We thank the reviewer for their insightful comment. We are very much interested in the work
of Fair et al 2023. The general lessons from their manuscript align with, and indeed partly
explain, our low-usage finding. Although we showed that colocalised splice junctions are
enriched for low-usage compared to high-usage junctions, the functional consequences of
these events are beyond the scope of our work. Fair et al. relied on multiple longitudinal
assays to capture the fate of transcripts along the RNA processing timeline:

*“Degradation was estimated as the difference in expression observed in steady-state RNA,*
*versus various measures of transcriptional activity, including promoter activity (H3K27ac)*
*and transcription elongation (H3K36me3)”*

Fair et al estimated the percentage of splice junctions that are uniquely present NMD-
predicted transcripts. However, this estimation was coupled with both nascent and steady-
state RNA-seq measurements, which enables them to validate their estimates. Similarly,
determining the fate of colocalised splice junctions (whether they are degraded or translated
to protein) would required the generation of either nascent RNA-seq data or mass
spectrometry data, which unfortunately is not possible in our dataset. We highlight a
potential role for low-usage splice junctions in complex disease risk, but we leave it to future
studies to determine their functional consequences in the context of complex disease risk.

**Changes to manuscript:**

We have added the following text referencing Fair et al. to our Discussion section:
*“Studies that examine the fate and role of low-usage splice junction in the context of complex*
*disease are set to improve our understanding of the function consequences of low-usage*
*splice junctions⁶².”*

**Minor concerns**

**- In general, it would great to see more visual examples of how the sQTLs look,**
**especially for response QTLs. For instance, for lines 138-143, it would be great to**
**have a figure showing this exon usage.**

We have now added RNA-seq coverage plots for all the examples that we detail in the main
or supplementary section of the manuscript, namely *PTPN2* (Figure 6d and Supplementary
Figure 19), *DENND1B* (Supplementary Figure 21) and *TOM1* (Supplementary Figure 22).

- Line 512: “RNA-seq libraries were produced for 217, which represented 209 lines
after quality control”. Is a word missing after “for 217”. Corrected
- Line 275: typo (figure 5c not 6c): Corrected
- Line 278: typo (figure 5b not 6b): Corrected
- Supplementary Figure S5 is odd, PTPN2 is shown for eQTL but not for sQTL:
Corrected (reference to eQTL and sQTL colocalisations was swapped in the caption). The
updated figure is now Supplementary Figure 18.
- Figure 6d : the legend is not explaining what are the blue/purple and yellow colors
(correspond to genotype of sLPS_6) and I would like to see the same plot for the
control condition: Added as Supplementary Figure 19
- Would be good to see the same plot as in Fig 6d for DENND1B and LRRK2 Added as
supplementary Figures 21 and 22 (*LRRK2* was replaced by *TOM1*; see “Additional
Corrections” for a justification)

Additional changes made in this revision

Title change

We have decided to change the title of our manuscript to “Splicing QTL mapping in
stimulated macrophages associates low-usage splice junctions with immune-mediated
disease risk”. The new title reflects our uncertainty around whether low-usage splice junction
underpin IMDs or whether they simply reflect decreased splicing efficiency that results in
increased erroneous splicing. Additionally, we have discussed this possibility in our
“Discussion” section.

Baseline conditions for defining response sQTLs

In our manuscript, we reported response sQTLs as sQTLs whose effect sizes were
significantly different upon stimulation. Ctrl_24 was used as a baseline condition against
which all other stimulation condition effect sizes were compared. As a result we excluded
genes which had response sQTLs in Ctrl_6 compared to the baseline Ctrl_24 as these were
likely false positives as both Ctrl_6 and Ctrl_24 are composed of resting macrophages.
Additionally, this may lead to an overestimation of response sQTLs in stimulation conditions
where RNA was harvested after 6 hours as these conditions were compared to an
unsuitable baseline condition. To avoid these biases, we have now changed the baseline
conditions as follows: Ctrl_24 for stimulation conditions where RNA was harvested 24 hours
following stimulation (e.g. IL4_24), and Ctrl_6 for stimulation conditions where RNA was
harvested 6 hours following stimulation (e.g. IL4_6). This has led to slightly different
numbers of response sQTLs in Figures 3b and 3c. For example, the number of sGenes with
response sQTLs in IL4_6 has decreased from 9% to 4%. This change has also impacted the
number of genes with a colocalised response sQTLs. Previously, we considered any genes
that exhibited evidence of response sQTLs between Ctrl_6 and the baseline condition
Ctrl_24 to not be response sQTLs. As we now compare each condition to its respective
timepoint, we have removed this requirement, resulting in an increase in the number of
genes with a colocalised response sQTL from 9.6% (68/707 genes) to 17.5% (124/707
genes).

Additionally, we found that the choice of Ctrl_24 as a baseline condition led to an
overestimate of the proportion of sGenes that harboured response sQTLs at 6 hours only
(Figure 3c). After changing the baseline conditions, we found that the number of genes with
6-hour-only or 24-hour-only response sQTLs depends on the condition.

Replacing the *LRRK2* example with *TOM1*

Regional association plots

Previously, we visualised our *PTPN2*, *LRRK2* and *DENND1B* associations via regional
association plots. In those plots, we *only* showed variants that were tested in both the GWAS
study under investigation and our sQTL data. Among those variants, we marked the variant
with the lowest P-value as the index variant. We have now amended these plots to show all
the variants tested in each study separately. This was particularly important for the *LRRK2*
association, where the lead SNP in the IBD GWAS study was not tested in our sQTL data as
its minor allele frequency was below MacroMap's MAF threshold (rs117981694;
12:40428296_G_A; MAF=2.9% in non-Finnish Europeans in 1000GP High Coverage;
MacroMap MAF threshold=5%). The shared SNP with the lowest P-value (rs76904798;
12:40220632_C_T; MAF=13.6% in non-Finnish Europeans in 1000GP High Coverage;
Figure) was not in LD with the lead IBD SNP, suggesting that it represents an independent
*LRRK2* signal in the region ($R^2=0.16$). We have decided to remove the *LRRK2* and instead
show an example where the primary disease signal colocalises with the sQTL.

Figure: Regional association plots between the IBD-associated signal and an LRRK2 signal. The lead IBD SNP (rs117981694) is not tested in MacroMap due to its low MAF (2.9% in non-Finnish Europeans in 1000GP high-coverage). The shared SNP with the lowest IBD P-value is indicated with an arrow (rs76904798). Colours indicate LD with the index SNP.

We then searched for other colocalisations that may implicate genes involved in endocytosis, similar to our hypothesis regarding LRRK2, which we outlined in our Results subsection “sQTL colocalisations converge on dysregulated pathways in IMDs”. We found a colocalisation between an ulcerative colitis (UC) locus at 22q12.3 and an sQTL for

*TOM1* (Figure). *TOM1* was shown to interact with several effector proteins at
 different stages of endosomal trafficking (reviewed in ref 13). We provide more
 details on this colocalisation in our Results subsection “sQTL colocalisations
 converge on dysregulated pathways in IMDs”, Figure 7 and Supplementary Figure
 22.

Figure: Regional association plots between the UC-associated signal and a *TOM1* signal. Colours indicate LD with the index SNP.

References

[1] Hukku, Abhay et al. "Probabilistic colocalization of genetic variants from complex and
molecular traits: promise and limitations." *American journal of human genetics* vol. 108,1
(2021): 25-35. doi:10.1016/j.ajhg.2020.11.012

[2] Joglekar, Anoushka et al. "Single-cell long-read sequencing-based mapping reveals
specialized splicing patterns in developing and adult mouse and human brain." *Nature*
*neuroscience* vol. 27,6 (2024): 1051-1063. doi:10.1038/s41593-024-01616-4

[3] Pardo-Palacios, F.J., Wang, D., Reese, F. et al. Systematic assessment of long-read
RNA-seq methods for transcript identification and quantification. *Nat Methods* (2024).
<https://doi.org/10.1038/s41592-024-02298-3>

[4] Chen, Qiuyue et al. "Genome-Wide Association Analyses Reveal the Importance of
Alternative Splicing in Diversifying Gene Function and Regulating Phenotypic Variation in
Maize." *The Plant cell* vol. 30,7 (2018): 1404-1423. doi:10.1105/tpc.18.00109

[5] Reimer, Thornik et al. "poly(I:C) and LPS induce distinct IRF3 and NF-kappaB signaling
during type-I IFN and TNF responses in human macrophages." *Journal of leukocyte*
*biology* vol. 83,5 (2008): 1249-57. doi:10.1189/jlb.0607412

[6] El-Zayat, S.R., Sibaii, H. & Mannaa, F.A. Toll-like receptors activation, signaling, and
targeting: an overview. *Bull Natl Res Cent* **43**, 187 (2019). [https://doi.org/10.1186/s42269-](https://doi.org/10.1186/s42269-019-0227-2)
[019-0227-2](https://doi.org/10.1186/s42269-019-0227-2)

[7] Komal, Asma et al. "TLR3 agonists: RGC100, ARNAX, and poly-IC: a comparative
review." *Immunologic research* vol. 69,4 (2021): 312-322. doi:10.1007/s12026-021-09203-6

[8] Lim, Chan Seok et al. "TLR3 forms a highly organized cluster when bound to a poly(I:C)
RNA ligand." *Nature communications* vol. 13,1 6876. 12 Nov. 2022, doi:10.1038/s41467-
[022-34602-0](https://doi.org/10.1038/s41467-022-34602-0)

[9] Alexopoulou, L et al. "Recognition of double-stranded RNA and activation of NF-kappaB
by Toll-like receptor 3." *Nature* vol. 413,6857 (2001): 732-8. doi:10.1038/35099560

[10] Marks, Michael S et al. "Lysosome-related organelles: unusual compartments become
mainstream." *Current opinion in cell biology* vol. 25,4 (2013): 495-505.
doi:10.1016/j.ceb.2013.04.008

[11] Manderson, Anthony P et al. "Subcompartments of the macrophage recycling
endosome direct the differential secretion of IL-6 and TNFalpha." *The Journal of cell*
*biology* vol. 178,1 (2007): 57-69. doi:10.1083/jcb.200612131

[12] Murray, Rachael Z et al. "Syntaxin 6 and Vti1b form a novel SNARE complex, which is
up-regulated in activated macrophages to facilitate exocytosis of tumor necrosis Factor-
alpha." *The Journal of biological chemistry* vol. 280,11 (2005): 10478-83.
doi:10.1074/jbc.M414420200

[13] Seimon, Tracie, and Ira Tabas. "Mechanisms and consequences of macrophage
apoptosis in atherosclerosis." *Journal of lipid research* vol. 50 Suppl,Suppl (2009): S382-7.
doi:10.1194/jlr.R800032-JLR200

[14] Ding, Fangyuan et al. "Dynamics and functional roles of splicing factor
autoregulation." *Cell reports* vol. 39,12 (2022): 110985. doi:10.1016/j.celrep.2022.110985

Response to referees
Manuscript: NCOMMS-23-33917A

Reviewers' comments:

**Reviewer #1 (Remarks to the Author):**

*I thank the authors for responding to my comments, which I think*
*improved the manuscript. I have several additional comments*
*regarding their responses.*

*1)"Rotival et al. have profiled the alternative splicing landscape of*
*monocytes exposed to four different stimuli, but have not robustly*
*linked sQTLs to disease-associated loci."*

*Rovital et al. have indeed linked sQTL to disease-associated loci.*
*Although I agree that the number of disease-associated loci is*
*increased in the present study, I still think the scientific advance*
*shown by the present study may not exceed that brought by*
*Rotival's study.*

We respectfully disagree with this comment. While it is true that both our work and that
of Rotival et al have both mapped sQTLs in an attempt to better understand disease
associated loci, we believe that our manuscript makes a number of major novel
contributions. Briefly, and without repeating our response to this comment from the
initial review, our manuscript:

- 1) We identified more than five times the number of sGenes compared to Rotival et
al. These discoveries are a scientific advance that more completely capture the
role of splicing regulation in iPSC-derived macrophages.
2) We identified more than 3.6 times the number of sQTL-disease colocalisation
(707 vs 195) compared to Rotival et al. This is a scientific advance that
nominates effector genes across a broad range of complex diseases.
Additionally, we relied on statistical colocalisation analysis rather than the
sQTL/GWAS SNP overlap method adopted by Rotival et al. Indeed, Rotival et al.
mention in their Discussion that "*Future work using colocalization analyses and*
*Mendelian randomisation approaches should help establishing a causal role of*
*AS in disease risk at the identified loci*".

3) We demonstrate that low-usage splice junctions comprise a significant
proportion of all sGene and an equivalent proportion of colocalising sQTLs. This
was not mentioned by Rotival et al and is a scientific advance that highlights the
importance of these junctions and the need for future studies to be designed in
such a way that these are captured. We also hope our work at least motivates
others to investigate whether genetically-controlled variation in low-usage splice
junction or a reduced usage of abundant splice junctions contribute to complex
disease risk. We believe that answering these questions is important to
understand the role of AS in complex disease risk. To this effect, we note that our
preprint has already been cited by Fair et al, Nature Genetics, 2024 for providing
this scientific advance.

2)"Our manuscript is submitted jointly with the MacroMap eQTL
manuscript to the same journal (Panousis et al. 2023)"

*I think this paper would be a split publication, but I would defer*
*this issue to the journal editors.*

Ideally, we would like the two papers to be published back-to-back because the sQTL
paper builds on the eQTL paper, but this is not a requirement. We are very happy for the
editor to decide if back-to-back publication is possible based on the respective
timelines through review.

**Reviewer #2 (Remarks to the Author):**

*Related to my original concern #2 and #5:*
*Because the major finding of this paper is related to low-usage*
*splicing junctions, it is absolutely necessary to carry out*
*experimental validations of the existence of such splicing events.*
*This reviewer is not asking the authors to test the hundreds of*
*hypotheses that your paper has generated (as you incorrectly*
*interpreted in your response). At the minimum, the authors needs*
*to experimentally test a random subset of the low usage splicing*
*junctions, in order to confirm their existence and rule out the*
*possibility of false positives.*

We thank the reviewer for their comment. We agree that our work would be
strengthened with further confirmation that the low-usage splice junctions identified in

our paper exist. To this end, we undertook additional quality control and validation
assays. As described in our response to your earlier comment, we performed a series of
QC checks to minimise false positives – including comparing the alignment score,
multimapping rate, and canonical versus non-canonical splice junction rate in low-
usage splice junctions versus more common usage splice junctions. Reassuringly,
these rates were consistent as outlined in Supplementary Figures 13-15 and our
section “Lowly-used alternative splicing events are associated with complex disease
risk”. Furthermore, we found that 99% of our splice junctions were reported in
Intropolis, a database of splice junctions detected by The Sequence Read Archive
following standardised analysis across over 20,000 short-read RNA-seq samples.

Spurred on by your comment, we considered approaches for experimentally validating
a subset of our low-usage splice junctions. We initially considered performing qPCR,
but this was not possible because, unfortunately, the decision was taken not to
preserve the ~ 4,700 MacroMap RNA samples. Furthermore, we did not preserve the
differentiated cell lines because they do not survive cryopreservation. In any case, a
qPCR approach would only allow us to test a small handful of splice junctions. Instead,
we sought an alternative approach that would efficiently allow us to attempt validation
across a large subset of colocalising splicing junctions. To this end, we focussed on 115
low-usage splice junctions that we colocalised with Crohn’s disease associated GWAS
loci (PP4 > 0.75 and mean IUR < 0.1).

To this end, we generated long-read single-cell RNA-seq data across terminal ileum
biopsies from 15 individuals (8 healthy individuals, and 7 Crohn’s disease patients). We
had previously generated short-read scRNA-seq data on these same samples and cells
(Krzak, Alegbe, Taylor & Jones. et al.; ref. 78 in the revised manuscript). We relied on
cell-type annotations that were generated based on the short-read RNA-seq data to
label cells based on eight broad cell-type categories (enterocytes, secretory cells, stem
cells, mesenchymal cells, T cells, B cells, Plasma B cells, and myeloid cells). We
processed the long-read data using the standard Iso-seq pipeline provided by PacBio,
mapped reads to the genome using minimap2 and identified transcripts using SQANTI3
(more details in the Methods section). Despite the small number of myeloid cells,
(N=1,203 from ~50,000 cells in total), we were able to replicate the existence of 58/115
splice junctions (50.4%) in this single cell type. Across all immune cells, we replicated
92/115 splice junctions (80%). We consider this an impressive validation rate given we
have only generated long-read data across 15 individuals (versus ~ 200 in MacroMap)
and have a relatively low number of myeloid cells. This gives us confidence that our
approach for calling splice junctions across MacroMap is robust. We will also release
data for the replicated splice junctions upon publication.

Changes to manuscript

We added the following text to subsection “Lowly-used alternative splicing events are
associated with complex disease risk”. In addition to this, we added Supplementary
Figure 19, which outlines the validation design, and Supplementary Figure 20, which
shows the number of samples and cell barcode for validated splice junctions. We have
also attached excerpts from the number of reads per sample, and the counts of each

validated splice junction per cell barcode as Supplementary Tables S4 and S5
respectively. The release of the full tables is expected upon publication when we obtain
appropriate release approvals:

*“To further assess the replication of a subset of our low-usage splice junctions, we*
*generated single-cell long-read RNA-seq data. We focussed our validation on a set of*
*115 low-usage splice junctions that we colocalised with Crohn’s disease-associated*
*signals (mean IUR < 0.1 and PP4 > 0.75). The long-read dataset consists of 15 terminal*
*ileum biopsies from eight healthy and seven non-inflamed CD individuals. We*
*previously identified major epithelial and immune cell types from the same samples via*
*short-read single cell RNA-seq⁷⁸. We therefore matched the cell barcodes from the*
*same samples to their cell-type annotation obtained from short-read RNA-seq to*
*identify myeloid cells (Supplementary Figure 19).*

*Full-length cDNA was sequenced using Pacbio’s concatenation method MAS-seq⁵⁸. We*
*generated a median of 96,347,033 segmented long-reads per sample, with a median*
*read length of 738 basepairs. After quality control (Methods), we obtained a median of*
*45,451,801 deduplicated long read UMIs per sample and a median of 2,627 and 4,324*
*cells per sample for healthy and CD samples respectively (Supplementary Table S4). As*
*the biopsies we used came largely from individuals with no inflammation, we were able*
*to identify only 1,203 myeloid cells across all 15 samples (Supplementary Figure).*
*Despite this small number, we validated the existence of 50.4% of CD-implicated low-*
*usage splice junctions in myeloid cells only (58 splice junctions). Replication rate*
*increased to 80% (92 splice junctions) when we performed the replication across all*
*immune cell types (B cells, T cells, Plasma B cells and Myeloid cells; 22,579 cells; full*
*list of all CD splice junctions and isoforms are provided as Supplementary Table S5 and*
*Supplementary Data). 80.4% of replicated splice junctions are detected in at least two*
*samples (N=74) and 85.9% of replicated splice junctions are detected in at least two*
*cell barcodes (N=79; Supplementary Figure 20). The replicated splice junction included*
*annotated splice junctions (N=49), splice junctions with an annotated donor only (N=17)*
*or acceptor only (N=12), or a novel combination of a known acceptor and a known*
*donor (N=14), suggesting that the replication reflects both annotated and unannotated*
*splice junctions (Supplementary Data).”*

**Original concern #3: regarding effect sizes,**
**The figure provided is not showing what its legend says**
**(specifically, y axis: absolute value of IUR). Based on the figure, it**
**appears that low-usage (black) splice junctions had smaller**
**absolute effect sizes than common-usage (orange) junctions. The**
**reviewer understands the argument made by the authors about**
**the normalization procedure. However, the effect size**
**comparison between low usage and common usage junctions is**

important. The authors needs to examine this difference, and
conclude whether low usage junctions are associated with
smaller effect sizes relatively. Such data and analysis should be
included in the manuscript.

We thank the reviewer for their comment. We formally investigated the difference
between the effect size distributions of common and low-usage splice junctions that
colocalised with disease-associated loci. We noted a small but significant difference
between the two distributions (mean=0.75 and 0.70 for common- and low-usage splice
junctions, respectively; Kruskal-Wallis P-value= 8.9×10^{-15}). We have added a sentence to
the manuscript highlighting this fact. We thank the reviewer once again for this
insightful suggestion.

*Figure: (Left) Cumulative distribution of absolute effect sizes for low-usage (black) and*
*common-usage (orange) splice junctions that colocalise with disease-associated loci*
*across all conditions(Right) boxplot showing the difference in absolute effect sizes*
*between common-usage and low-usage splice junction. Common- and low-usage*
*splice junctions exhibit a small but significant difference in mean absolute effect sizes*
*(mean absolute effect sizes=0.7 and 0.75 and standard deviation=0.26 and 0.27,*
*respectively; Kruskal-Wallis test P-value= 8.9×10^{-15}).*

**Changes to manuscript**

We have acknowledged this observation in subsection “Lowly-used alternative splicing
events are associated with complex disease risk” and in the figure below as
Supplementary Figure 12:

*“Generally, we noted that there was a small but significant difference in absolute effect size
between common- and low-usage splice junctions that colocalised with disease associated
loci (mean effect size=0.75 and 0.7 and standard deviation=0.27 and 0.26 respectively;
Supplementary Figure 12).”*

*Original concern #4: “For each intron, are all genotypes
associated with low IUR, or certain genotypes with relatively high
IUR (e.g. >0.1) and others with low values? If it is the latter case,
then it does not make much sense to call them lowly used.” The
authors did not understand or interpret my question correctly. In
their response, the authors state “A mean intron usage ratio < 0.1
indicates that across all individuals the mean intron usage is less
than 0.1. As this is a mean across all individuals in the study, we
believe it is appropriate to refer to them as low usage intron
rations.” This definition of “low-usage” splice junctions may be
misleading because it is only based on a mean value. To be
identified as significant sQTL events, the so called “low-usage”
splice junctions should have genotypes that are NOT “low-usage”
as sQTL looks for “a significant difference in mean intron usage
with respect to genotype” (stated by the authors). Thus, calling
these events “low-usage” based on the mean is misleading,
which introduces an artificial sense of novelty to the manuscript.*

We thank the reviewer for their comment. We would like to clarify that when a splice
junction has a significant usage-increasing sQTL, that does not necessarily mean that
by definition, it will have a genotype group with a mean intron usage ratio (IUR) > 0.1.
Although this could happen for a few splice junctions, this depends on the overall mean
IUR and the effect size of the associated variant. For common variants, effect sizes are
unlikely to be very large.

We further investigated this in our set of colocalised low-usage splice junctions. We
found that 19.5% of intron-condition pairs have have a mean IUR > 0.1 in the genotype
group with 2 copies of the usage-increasing allele. At the level of individual conditions,
this percentage ranges from 13.3%-28.4%, and 1.2%-9.3% have mean IUR > 0.2 in the
genotype group with the two copies of usage-increasing alleles (Figure). Given that
approximately half of colocalised splice junctions are considered low-usage under our
current definition, adopting the alternative definition of low-usage splice junctions will

lead to a modest decrease in the overall percentage of low-usage splice junctions. We
 are happy to switch to the alternative definition if it is deemed necessary by the
 reviewer/editor. Since this alternative definition excludes approximately one fifth of
 colocalised low-usage intron-condition pairs, we do not think that it will significantly
 undermine the finding regarding low-usage splice junctions.

*Figure: (Top) Density plot of mean IURs in individuals with 0, 1 or 2 copies of usage-*
 *increasing allele (x-axis) with the density of intron-condition pairs shown on the y-axis*
 *(Bottom) Proportion of colocalised low-usage splice junction that have a mean IUR <*
 *0.1, IUR ≥ 0.1 and IUR ≥ 0.2 in the genotype group with two copies of the usage*
 *increasing allele.*

Reviewer #3 (Remarks to the Author):

The authors have substantially improved their manuscript and they have address my comments. My only remaining minor requests are the following:

- I hope the authors have given more experimental methods details of the stimulation conditions (including concentration of each stimulant used) in the accompanying Panousis et al. manuscript, but if not, please include those details in the methods section of this manuscript.

We thank the reviewer for their comment. We have given detailed experimental details in Panousis et al. including concentrations of stimulants, which combinations were used in some conditions, the detailed macrophage differentiation protocol used and macrophage purity assay used to ensure the quality of differentiated cells, RNA-seq preparation and and quality control (see Supplementary Note in Panousis et al 2023 on biorXiv).

- The sQTL visualization plots that the authors have generated (such as in Fig. 6d) are really helpful to appreciate the events behind the sQTL co-localizing with disease risk loci. If the authors could share their code for creating those plots in their GitHub page, it would be greatly appreciated by the sQTL community (and it could increase their number of citations), especially given the LeafViz app does not allow visualizing splicing effects for more than 2 groups.

Reviewer #3 (Remarks on code availability):

I went into their github and saw their reported scripts. I didn't go deep enough to know whether the info that they provide is enough for replicating their main results. However, as I mentioned to the authors, if they make the code for visualizing sQTL available I believe it would be widely used.

We thank the reviewer for their comment and we are pleased that they find it useful. We have relied on a tool developed by Mu et al. (github.com/Zepeng-Mu/pyGenomeTracks), but we have also made our wrapper code available and documented on github (github.com/andersonlab/macromapsqtl/tree/main/sashimi). We have cited Mu et al. in other parts of the manuscript, but we also explicitly linked the Github repository.

Changes to manuscript

We added the following as a subsection of the Methods section with the title “RNA-seq
coverage plots”:

“RNA-seq coverage plots were generated using custom scripts based on
pyGenomeTracks⁷⁵ with an additional sashimiBigWig track from Mu et al^{45,76}. Wrapper
scripts are available at github.com/andersonlab/macromapsqtl/tree/main/sashimi.
Briefly, RNA-seq coverage and intron usage ratios were averaged within each genotype
group for a given variant using the commands samtools view and samtools merge. RNA-
seq coverage was then calculated using the command bamCoverage using a binsize of
10 and output as a bedgraph file. Bedgraph files were then converted to bigWig files
using the command bedGraphToBigWig⁷⁷ which were then visualised using
pyGenomeTracks.”

Response to referees

Manuscript: NCOMMS-23-33917C

Reviewer#2

“Regarding the original concern #4: I understand the authors’ response, and it addressed my question. However, to enhance scientific rigor, the authors should re-define “low usage” splice junctions. Instead of defining them based on a mean value, such junctions should be defined requiring all genotypes had $IUR < 0.1$. This is important because it is not correct to call splice junctions “low usage” if they are not lowly used in certain people. The fact that such junctions had varying IUR levels across genotypes, where some $IUR > 0.1$, makes it interesting.”

We would to thank the reviewer for their suggestions. We have changed the definition of low-usage IUR to splice junctions that have an $IUR < 0.1$ in all genotype groups. Accordingly, we have reproduced Figure 4 and Supplementary Figures 12-20. We have also changed the relevant numbers and percentages in section “Lowly-used alternative splicing events are associated with complex disease risk”.